# The deubiquitylating enzyme Fat facets promotes Fat signalling and restricts tissue growth

Lauren E. Dawson [1,2], Aashika Sekar [1,3,5], Alexander D. Fulford [1,4,5], Rachel I. Lambert [1], Hannah S. Burgess [1] & Paulo S. Ribeiro [1] ✉

Tissue growth is regulated by many signals, including polarity cues. The Hippo signalling pathway restricts tissue growth and receives inputs from the planar cell polarity-controlling Fat signalling pathway. The atypical cadherin Fat restricts growth via several mechanisms that ultimately control the activity of the pro-growth transcriptional co-activator Yorkie. Fat signalling activates the Yorkie inhibitory kinase Warts, and modulates the function of the FERM protein Expanded, which promotes Hippo signalling and also directly inhibits Yorkie. Although several Fat pathway activity modulators are known to be involved in ubiquitylation, the role of this post-translational modification in the pathway remains unclear. Moreover, no deubiquitylating enzymes have been described in this pathway. Here, using in vivo RNAi screening, we identify the deubiquitylating enzyme Fat facets as a positive regulator of Fat signalling with roles in tissue growth control. Fat facets interacts genetically and physically with Fat signalling components and regulates Yorkie target gene expression. Thus, we uncover a role for reversible ubiquitylation in the control of Fat signalling and tissue growth regulation.

Developmental tissue growth and morphogenesis are controlled by a plethora of molecular mechanisms that must be tightly regulated to achieve reproducible organ and body size. In epithelia, one of the most important pathways involved in tissue growth regulation is the Hippo (Hpo) pathway, an evolutionarily conserved signalling cascade that integrates multiple signals that report on epithelial integrity[1–4]. Hpo signalling culminates in the inhibition of Yorkie (Yki; mammalian YAP), a transcriptional co-activator that associates with transcription factors such as Scalloped (Sd; mammalian TEAD1-4) to promote the expression of genes involved in cell proliferation and inhibition of apoptosis[1,5,6]. Yki activity is restrained by a kinase cascade consisting of the kinases Hpo and Warts (Wts; mammalian LATS1/2). The latter directly phosphorylates Yki, inhibiting its nuclear translocation, primarily by promoting interaction with 14-3-3 proteins[5,7]. Given its crucial

role in tissue growth control, homoeostasis is maintained by tight regulation of Hpo signalling, including a negative feedback loop in which Yki/YAP promote the expression of upstream activators of the kinase cascade[8–10].

Among the signals that regulate Hpo signalling are inputs from the cellular polarity machinery[3,4,9]. Hpo signalling is regulated both by proteins involved in establishing and maintaining apico-basal polarity (e.g., Crumbs (Crb), Scribble (Scrib), among others)[4,9], and planar cell polarity (PCP), such as the members of the Fat (Ft; mammalian FAT1-4) signalling pathway[11,12]. Ft is an atypical cadherin that localises to the sub-apical domain of epithelial cells and forms an heterotypic adhesion complex with the atypical cadherin Dachsous (Ds; mammalian DCHS1/2)[11], which is regulated by the Golgi resident kinase Four-jointed (Fj; mammalian FJX1)[13]. The combination of the opposing Ds

[1]Centre for Tumour Biology, Barts Cancer Institute, Queen Mary University of London, Charterhouse Square, London, UK. [2]The CRUK Gene Function Laboratory and Breast Now Toby Robins Research Centre, The Institute of Cancer Research, Fulham Road, London, UK. [3]Apoptosis and Proliferation Control Laboratory, The Francis Crick Institute, 1 Midland Road, London, UK. [4]Department of Developmental Biology, Washington University School of Medicine, St. Louis, MO, USA. [5]These authors contributed equally: Aashika Sekar, Alexander D. Fulford. ✉e-mail: p.baptista-ribeiro@qmul.ac.uk

and Fj expression patterns in tissues, and the differential effect of Fj-mediated phosphorylation on the affinity of Ft and Ds to each other results in a gradient of Ft signalling that contributes to the regulation of PCP and tissue growth[11,14–16].

Ft-mediated regulation of growth involves several mechanisms. Ft inhibits the function of the atypical myosin Dachs (D)[17–19], a known negative regulator of Wts function, albeit the precise molecular mechanisms remain unclear[17,20,21]. Ft also limits the activity of the zDHHC9-like transmembrane palmitoyltransferase Approximated (App)[22,23] and the D-interacting protein Dachs ligand with SH3s (Dlish), which control D sub-cellular localisation and function[24,25]. Moreover, Dlish also inhibits Hpo signalling in a D-independent manner, via the regulation of the upstream activator Expanded (Ex)[26]. Dlish interacts with Ex and promotes its degradation via the recruitment of Skp-Cullin-F-box (SCF) E3 ubiquitin ligase complexes containing the F-box protein Slimb, a known regulator of Ex function[27–29].

Interestingly, besides Dlish-mediated regulation of Ex stability, several steps of the Ft signalling pathway appear to be regulated by post-translational modifications such as ubiquitylation. D is thought to regulate Wts protein levels by an unknown mechanism that is likely to involve ubiquitylation[17,20]. In addition, Ft-mediated regulation of D function is at least partly dependent on the F-box protein Fbxl7, though it is still unclear whether D itself is ubiquitylated and degraded[30,31]. Finally, a recent report identified the E3 ligase Early girl (Elgi) as a new Ft signalling component involved in tissue growth regulation[32]. Elgi is a D-interacting protein that controls D protein levels and, along with App, is proposed to control D and Dlish localisation to the apical membrane[32]. Despite these observations, the precise molecular mechanisms by which ubiquitylation regulates Ft signalling and, by extension, tissue growth remain incompletely characterised. Importantly, to date there have been no reports of deubiquitylating enzymes (DUBs) as potential regulators of Ft signalling components.

To address this, we performed an in vivo RNAi modifier screen to uncover DUBs involved in Ft signalling and identified Fat facets (Faf; mammalian USP9X) as a regulator of tissue growth. In *Drosophila*, *faf* has not previously been connected to Hippo signalling. However, various studies have shown that Faf regulates *Drosophila* eye and embryonic development via the deubiquitylation and/or stabilisation of targets such as Liquid facets (Lqf)[33], D-Jun[34], Medea[35], and Dscam1[36]. Here, we show that Faf genetically and physically interacts with Ft signalling components, and controls expression of Yki target genes. Therefore, the function of Faf illustrates the crucial role of ubiquitylation in the regulation of Ft signalling and tissue growth.

## Results

### Identification of Faf as a deubiquitylating enzyme involved in Ft signalling

Previous studies have identified E3 ubiquitin ligases involved in the regulation of Ft signalling, such as Elgi and a Fbxl7-containing SCF complex[30–32]. However, to date, a role for deubiquitylating enzymes (DUBs) in Ft signalling regulation has not been described. To identify DUBs that modulate Ft function, we performed an in vivo RNAi modifier screen using the *Drosophila* adult wing as a model (Figure S1a). We used the wing driver *nub-Gal4* (*nub >*) to target the full complement of *Drosophila* DUBs with *UAS-RNAi* transgenes. Each *DUB^RNAi* line was crossed to *nub-Gal4, UAS-ft* (*nub > ft*) or *nub-Gal4* (*nub >*) as a control. Figures S1b and S1c show the in vivo screening results and quantification of the relative wing size of the different genotypes tested. In agreement with its effect on Hpo and PCP signalling, *UAS-ft* expression in the wing pouch resulted in smaller and rounder wings (Figs. 1d, j, k and s2d) compared with controls (Figs. 1a, j, k, S1d, S2a, S2c and S2d).

As a result of our screening approach, we identified the DUB Fat facets (Faf, encoded by *faf, CG1945*) as a potential regulator of Ft

signalling. *faf* depletion using several independent RNAi lines resulted in a partial suppression of the Ft undergrowth phenotype (Figs. 1e, j and S1c). Interestingly, Faf seems to primarily affect the tissue growth function of Ft, but not its PCP function, based on the ratio of the lengths of the anterior-posterior (AP) and proximal-distal (PD) axes of the adult wing, wing circularity and the orientation of adult wing hairs (Figs. 1k, S2b-d and S2i; see Materials and Methods for details). Next, we sought to determine if modulation of *faf* expression alone affects tissue growth and to validate its interaction with Ft signalling. *faf* depletion in the developing wing using *nub-Gal4* resulted in a mild increase in wing size, when compared to controls (Figs. 1j and S1e–g). Conversely, over-expression of *faf* (Faf^isoform C^, *faf^isoC37^*; Fig. 2a) reduced wing size (Figs. 1c, j and S1h). When combined with *UAS-ft*, depletion of *faf* suppressed the Ft phenotype (Fig. 1e and j), while *faf* over-expression enhanced it (Fig. 1f and j). In contrast, simultaneous depletion of *ft* and *faf* caused an enhancement of the *ft^RNAi* phenotype (Fig. 1g, h and j), while over-expression of *faf* in the context of *ft^RNAi* partially suppressed the wing overgrowth phenotype (Fig. 1g, i and j). Interestingly, we observed that modulation of Faf levels resulted in a very specific defect of the L2 wing vein, with the appearance of extra wing material in *faf^RNAi* wings (Figs. 1b and S2f-h). Notably, this phenotype has previously been associated with changes in Ft levels[38,39] and was also observed when Ft levels were modulated in isolation (Fig. 1d and g). Importantly, modulating Faf levels enhanced the L2 wing vein phenotypes (Figure S2h). Our results suggest that Faf promotes Ft activity in tissue growth control.

To validate the results obtained with *faf^RNAi*, we next assessed if *faf* genetically interacts with *ft* mutations in the regulation of tissue growth. Due to its critical role in controlling Hpo signalling, *ft* homozygous mutations are lethal and associated with extreme tissue overgrowth phenotypes[40,41]. However, certain *ft* mutations allow the analysis of tissues from late L3 larvae in a trans-heterozygous situation, such as the *ft^G-rv^/ft^8^* combination, allowing them to be studied alongside other genetic alterations[15,42]. *ft^G-rv^/ft^8^* trans-heterozygous mutant wing discs displayed extreme tissue overgrowth, compared to wild-type (*w^iso^*) wing discs (~170% larger than controls; compare Fig. 2b and c). Remarkably, when we combined *ft^G-rv^/ft^8^* mutations with various *faf* mutant alleles (Fig. 2a), we observed an enhancement of the tissue overgrowth phenotype (Fig. 2d-h). This indicates that *faf* genetically interacts with *ft* and that Faf function is important to restrict tissue growth. Given that *faf* mutation enhances *ft* mutant phenotypes, it is possible that Faf controls growth in both Ft-dependent and Ft-independent manners. Alternatively, Faf may act on residual Ft protein in *ft^G-rv^/ft^8^* trans-heterozygotes. We assessed this latter possibility by monitoring Ft protein levels in wing discs from control flies (Figure S3a), *ft^G-rv^/ft^8^* trans-heterozygotes (Figure S3b) and *ft^G-rv^/ft^8^* trans-heterozygotes with loss of a copy of *faf* (*faf^B3^*, Figure S3c). *ft* mutation leads to a dramatic reduction in Ft protein levels and, in the absence of *faf*, Ft localisation at the cell periphery appears further reduced. Taken together, our data in the *Drosophila* wing are consistent with Faf promoting the tissue growth suppressing function of Ft.

### Faf genetically interacts with core Ft signalling proteins

Having observed that Faf genetically interacts with Ft and plays an important role in the regulation of tissue growth, we extended our analysis to other members of the Ft signalling pathway that directly interact with Ft, such as Ds, Dlish and Fbxl7 (Fig. 3). Ft and Ds regulate tissue growth at least in part via their physical interaction across cell boundaries[11]. This interaction enhances the activity of Ft, thereby promoting its growth-suppressing function[11]. Accordingly, we found that Ds over-expression caused a reduction in wing size (Fig. 3b and p). Similarly to what was observed with Ft, *faf^RNAi* reversed this Ds-induced phenotype (Fig. 3c–e and p). This is consistent with a positive role for Faf in Ft signalling.

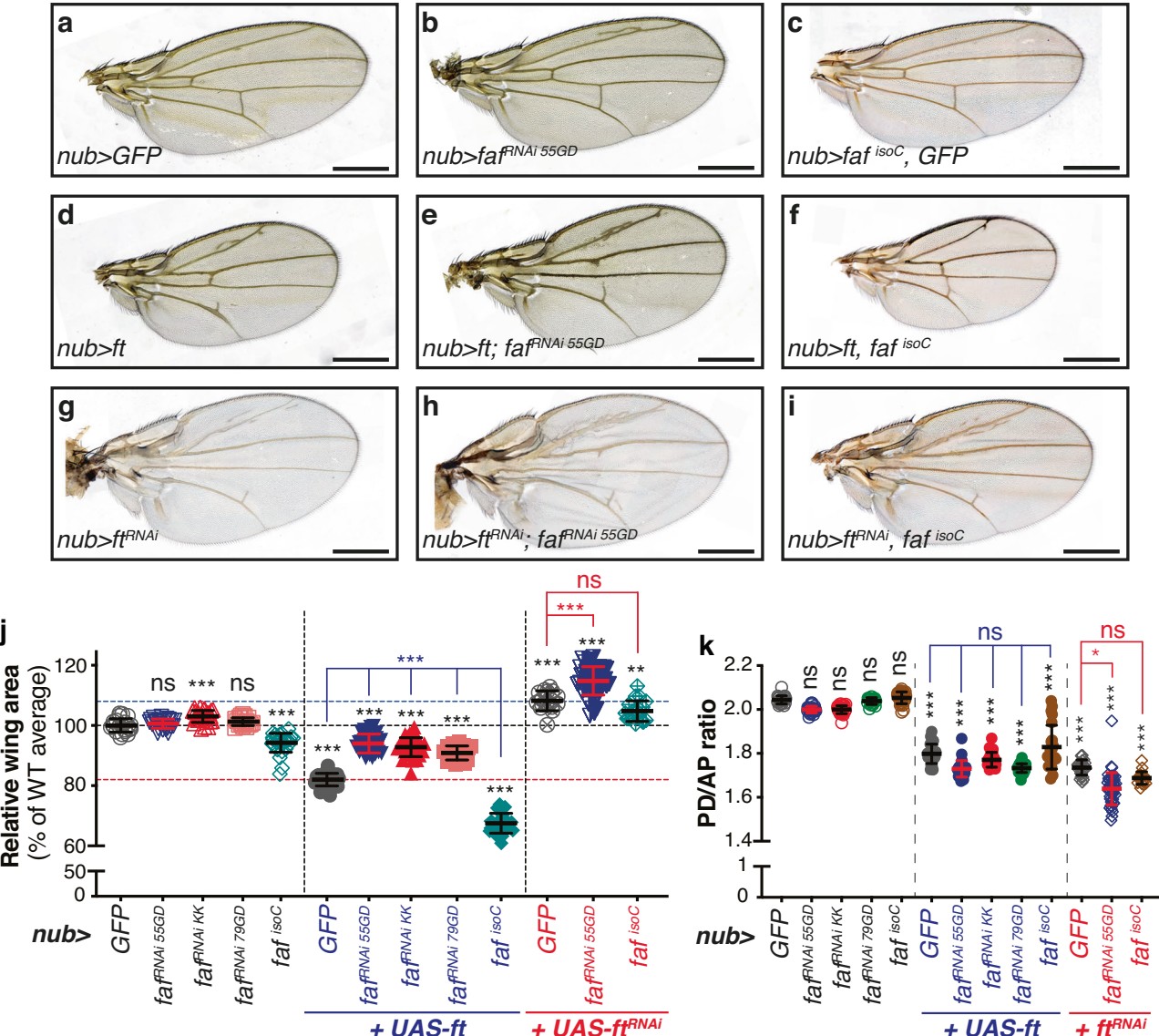

**Fig. 1 | Faf modulates Ft-mediated regulation of tissue growth. a–i** Modulation of Faf expression affects tissue growth during normal development or in conditions when Ft is over-expressed or depleted. Shown are adult wings from flies raised at 25 °C expressing the indicated transgenes in the wing pouch under the control of *nub-Gal4* (*nub >*). Compared to control adult wings expressing GFP (**a**), Ft-expressing wings were smaller (**d**), while *ft* depletion caused increased growth (**g**). Depletion of *faf* mildly enhanced tissue growth (**b**), while *faf* over-expression resulted in undergrowth (**c**). *faf* depletion resulted in a partial rescue of the undergrowth phenotype caused by *UAS-ft* (**e**), while it enhanced the overgrowth phenotype of *ft*^RNAi^ flies (**h**). In contrast, *UAS-faf* enhanced the growth impairment of *UAS-ft* flies (**f**) and mildly suppressed the overgrowth phenotype of *ft*^RNAi^ flies (**i**). **j** Quantification of relative adult wing sizes from flies expressing the indicated transgenes under the control of *nub-Gal4*. Data are represented as % of the average wing area of the respective controls (*nub > GFP*, average set to 100%). Data are shown as average ± standard deviation, with all data points depicted. ($n = 22, 25, 29, 20, 31, 26, 27, 29, 25, 23, 15, 47$ and $19$). Significance was assessed using Brown-Forsythe and Welch one-way ANOVA analysis comparing all genotypes to their respective controls (*nub > GFP*, *nub > ft* or *nub > ft*^RNAi^; black, blue or red asterisks, respectively) with Dunnett's multiple comparisons test. *, $p < 0.05$; **, $p < 0.01$; ***, $p < 0.001$; ns, non-significant. Scale bar: 500 µm. **k** Quantification of wing shape. Data is represented as the ratio between the length of the PD axis and the length of the AP axis. All data is represented as average ± standard deviation, with all data points depicted. Vertical dashed lines separate different genotype conditions (*nub >*, *nub > ft* or *nub > ft*^RNAi^). Significance was assessed using Kruskal-Wallis ANOVA analyses comparing all genotypes to the respective control (*nub > GFP*, *nub>ft* or *nub>ft*^RNAi^; black, blue or red asterisks, respectively), with Dunn's multiple comparisons test. *, $p < 0.05$; **, $p < 0.01$; ***, $p < 0.001$. ns, non-significant. ($n = 22, 25, 29, 20, 31, 26, 27, 29, 25, 23, 25, 47$ and $19$).

We also assessed if Faf could modulate phenotypes caused by downstream effectors of Ft signalling, such as Dlish and Fbxl7[24,25,30,31]. Dlish negatively regulates Hippo signalling and, therefore, promotes tissue growth[24–26]. Accordingly, depletion of Dlish (*Dlish*^RNAi^) in the developing wing resulted in reduced tissue growth (Fig. 3g and q). Co-depletion of *faf* and *Dlish* suppressed this phenotype (Fig. 3h and q), whilst Faf over-expression enhanced the undergrowth (Fig. 3i and q). As Dlish function is influenced by Ft, we combined *Dlish*^RNAi^ with Ft over-expression, which resulted in an enhancement of the *Dlish*^RNAi^-induced undergrowth (Fig. 3j and q). To address if this is dependent on Faf activity, we co-depleted *faf* in these conditions (*Dlish*^RNAi^ + *ft* over-expression), which led to a rescue of the phenotype (Fig. 3k and q), suggesting that Faf does indeed modulate Dlish phenotypes via Ft. As previously observed, *faf*^RNAi^ mostly affected wing size rather than shape, as wings remained rounder than controls (Fig. 3k), further indicating that Faf has a minor role in Ft-mediated regulation of tissue shape.

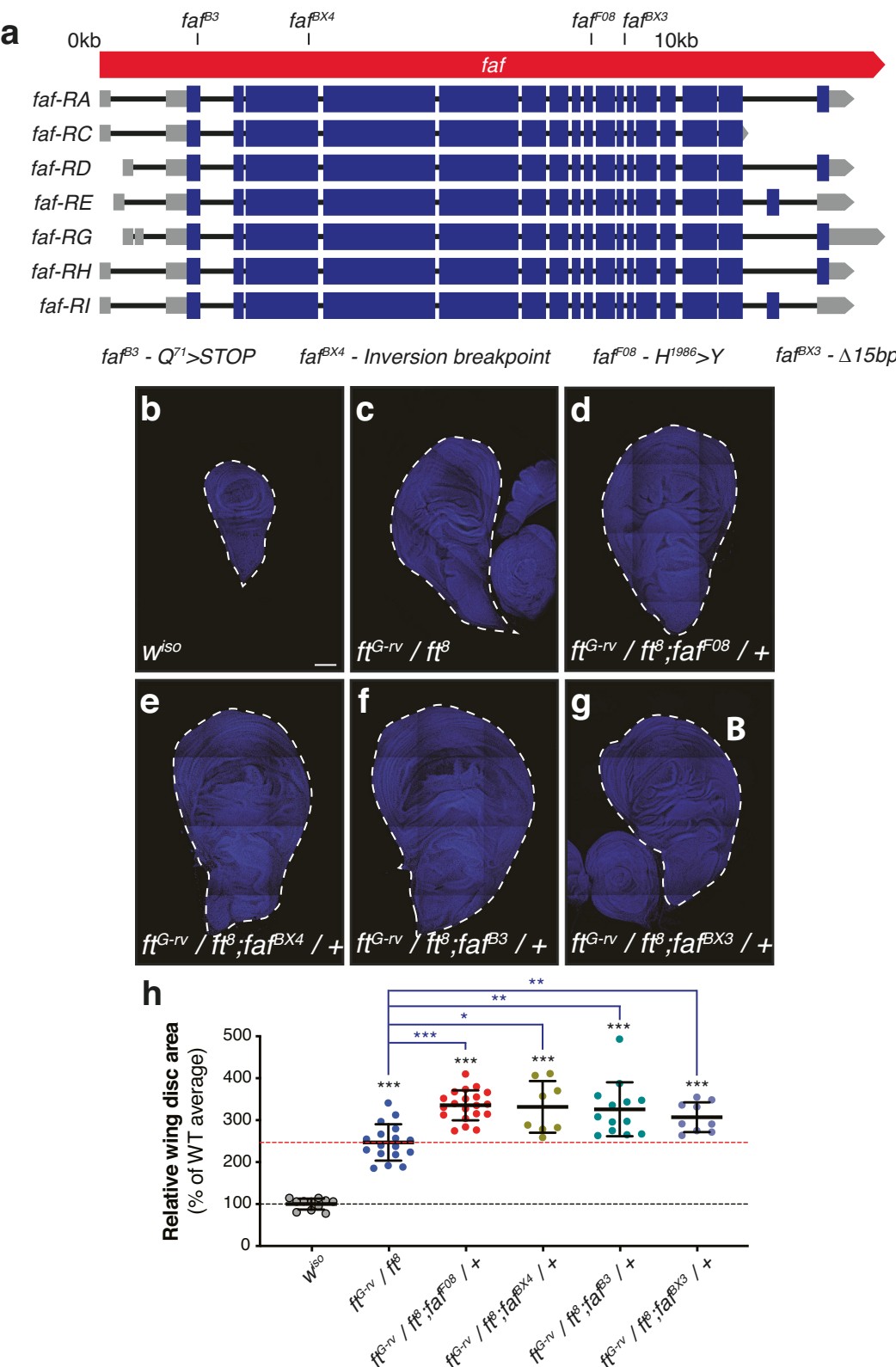

In agreement with its previously reported role, over-expression of Fbxl7 leads to a significant reduction in tissue size[30,31] (Fig. 3l and r). Interestingly, depletion of *faf* did not significantly affect the Fbxl7 phenotype, suggesting that Faf may act at the level of Fbxl7 (Fig. 3m and r). Moreover, Ft over-expression failed to enhance the Fbxl7 undergrowth phenotype (Fig. 3n and r), which is consistent with Fbxl7

playing a crucial role downstream of Ft. *faf*[RNAi] only had a slight effect on tissue growth in these conditions (Fig. 3o and r). We then extended our observations to additional Ft-associated proteins, such as Ex, Elgi and App (Figure S4). Our results suggest that Faf modifies phenotypes associated with Ex (Figure S4b-e and S4p) and Elgi (Figure S4f-j and S3q), but not with App (Figure S3k-o and S3r). Taken together, these

**Fig. 2 | *faf* genetically interacts with *ft*. a** Schematic representation of the *faf* locus. mRNA transcripts encoded in the *faf* locus are represented (boxes depict exons, while introns are shown as black lines. Coding sequences are shown as blue boxes, and UTRs are represented as grey boxes). Mutant alleles used are mapped onto the *faf* locus. *faf^B3* corresponds to a nonsense mutation leading to a truncated Faf protein (Q71 > STOP). The *faf^BX4* lesion is the site of an inversion breakpoint. *faf^F08* is a point mutation in the DUB catalytic domain (H1986 > Y). *faf^BX3* corresponds to a 15 bp deletion in the vicinity of an exon-intron boundary. **b–g** Loss of one copy of *faf* enhances the overgrowth of *ft* mutants. Shown are tiled confocal images of wing imaginal discs. Compared to control L3 imaginal discs (*w^iso*, b), the trans-heterozygous combination of *ft^G-rv* and *ft^8* leads to a dramatic increase in tissue size

(c), which is further enhanced by the presence of a *faf* mutant allele (*faf^F08* (**d**), *faf^BX4* (**e**), *faf^B3* (**f**), *faf^BX3* (**g**)). **h** Quantification of relative wing imaginal disc size. Data are represented as % of the average imaginal wing disc area of controls (*w^iso*, which was set to 100% and indicated by the black dashed line). Red dashed line indicates average wing disc size of *ft* trans-heterozygous mutants. Data are shown as average ± standard deviation, with all data points depicted. (*n* = 11, 17, 20, 8, 13 and 9). Significance was assessed using Brown-Forsythe and Welch one-way ANOVA analyses comparing all genotypes to the respective control (*w^iso* or *ft^G-rv*/*ft^8*; black or blue asterisks, respectively), with Dunnett's multiple comparisons test. *, $p < 0.05$; **, $p < 0.01$; ***, $p < 0.001$. Scale bar: 100 μm.

results suggest that Faf genetically interacts with multiple Ft signalling components and may work at the level of Fbxl7 and App, and potentially antagonistically to Dlish.

To position Faf within the Hpo pathway, we additionally tested whether modulation of Faf levels could enhance or suppress phenotypes elicited by the Hpo and Wts kinases. We combined Faf expression and *faf^RNAi* with depletion of *hpo* (*hpo^RNAi*, Figure S5a–d) or over-expression of *wts* (Figure S5e–k). Faf did not alter the *hpo^RNAi* over-growth phenotype in the adult wing, suggesting that Hpo acts downstream of Faf (Figure S5d). Additionally, while expression of *faf* resulted in an enhancement of the undergrowth phenotype elicited by *wts* expression, *faf* depletion had no effect, suggesting that Faf acts upstream of Wts (Figure S5k), in agreement with our observations regarding the Ft signalling components.

## Faf physically interacts with Ft and regulates its protein levels

Next, we assessed whether the genetic interactions between *faf* and members of the Ft signalling pathway could be the result of specific protein-protein interactions. For this, we expressed Faf in *Drosophila* S2 cells (Fig. 4a; Faf^LD, a protein encoded by the cDNA clone *LD22582*) and performed co-immunoprecipitation (co-IP) assays with the intracellular region of Ft (Ft^ICD43). Ft^ICD was readily detected in Faf^LD co-IPs, but not in the respective GFP controls (Figs. 4b and S6a). In agreement with the widespread role of DUBs as regulators of protein stability, we noticed that the protein levels of Ft^ICD in cell lysates appeared higher when Ft^ICD was co-expressed with Faf^LD (Fig. 4b). Therefore, we conducted further experiments using Faf^LD expression or *faf* RNAi-mediated depletion to validate this observation. Indeed, Ft^ICD was stabilised in the presence of Faf^LD (Fig. 4c and S6b) and, conversely, Ft^ICD levels were reduced when endogenous *faf* was depleted from S2 cells (Fig. 4d, S6c and S6d). This suggests that Faf may regulate Hippo signalling and tissue growth by modulating Ft protein levels.

To validate these observations in vivo, we modulated Ft or Faf expression in the posterior compartment of the wing disc (using *hh-Gal4*) and assessed the effect on Ft protein levels using a specific antibody[19,43,44] (Fig. 4h). As a control for general effects on the levels of proteins localised at the apical cell surface, we monitored Armadillo (Arm; *Drosophila* β-catenin) protein levels. Altering Faf levels in vivo recapitulated the results observed in S2 cells, and *faf^RNAi* expression resulted in a reduction in Ft levels (Figs. 4e, f, S7a, S7b, S7d and S7e), whilst over-expression of Faf increased Ft protein levels (Figs. 4e, g and S7d and S7e). Importantly, the effect of Faf on Ft levels was specific as Arm levels were generally unaffected (Fig. 4e′-h′, Figure S7c). Using *mirr-Gal4* to control gene expression, we also observed that Ft levels were regulated by Faf in the developing eye imaginal disc (Figure S7f–i). Expression of *faf* resulted in increased levels of Ft as assessed using the Ft-specific antibody (Figure S7f and S7h). In this tissue, Ft levels appear to be more susceptible to increased levels of Faf, as *faf* depletion had no effect (Figure S7f and S7i). Together, these data suggest that Faf regulates Ft protein levels both in *Drosophila* S2 cells and in vivo.

## Effect of Faf on Dachs subcellular localisation

Next, we tested whether Faf regulates events downstream of Ft and focused on a potential modulation of D function. As extensively documented[15,17,18], D is one of the most important effectors of Ft signalling[15,17,18], and Ft regulates D by controlling its subcellular localisation[18]. To assess D localisation, we used a *D::GFP* knock-in allele in combination with *en-Gal4, UAS-RFP*. This allowed us to monitor D subcellular localisation in the posterior compartment of the developing wing disc (marked by RFP) and use the anterior compartment as a control (Figure S8a). Additionally, Arm was used as a control for global effects on apical protein localisation (Figure S8h-m). As seen in Figure S8b, in control flies (*lacZ^RNAi*), D is localised at the membrane, similarly to Arm, and the stereotypical D polarisation toward the distal side of the cell can be observed with no major differences seen between the anterior and posterior compartments. We also assessed the effects on D localisation by monitoring the plot profile of both D and Arm (Figure S8b″). In agreement with published data[18,20,22,30,31,45,46], over-expression of Ft resulted in D mislocalisation, which appeared both more cytoplasmic and less polarised at the membrane, compared to controls (Figure S8c). In agreement with our previous results, combining Ft over-expression with *faf* depletion (*UAS-ft, UAS-faf^RNAi*) resulted in a return to control conditions (Figure S8d). In contrast, *faf^isoC* expression resulted in increased levels of cytoplasmic D and, in some cases, we observed patches of tissue where D was not localised at the membrane, a phenotype similar to Ft over-expression (Figure S8e)[18]. Depletion of *ft* or *faf* resulted in less apparent D polarisation (Figure S8f and S8g). Importantly, no changes were observed at the level of Arm localisation in any condition (Figure S8h-m). Together, these results suggest that Faf is, at least partly, required for regulation of signalling events downstream of Ft.

## Faf regulates Yki target genes in vivo

Our data suggest that Faf regulates tissue growth by modulating Ft function. To confirm that the effects of Faf on tissue growth were indeed due to changes in signalling activity downstream of Ft, we tested whether Faf could affect gene expression modulated by Yki, the main effector protein regulated by the Hippo pathway. For this, we monitored Yki-mediated transcription using as readouts two widely used Hippo signalling in vivo reporters; *ex-lacZ*[47] (Fig. 5a-f) and *HRE-diap1::GFP*[48] (Fig. 5g-l). Transgenes were specifically expressed in the posterior compartment of the wing using *en-Gal4* and marked by the expression of GFP or RFP, respectively in *ex-lacZ* or *DIAP1::GFP* experiments. As a control, we used *hpo^RNAi*, which is known to lead to increased Yki-mediated gene expression[7,49] (Fig. 5b, f, h and l). In agreement with our observations regarding the effect of Faf on tissue growth, *faf^RNAi* caused an increase in Yki-mediated transcription, consistent with decreased Hippo signalling activity (Fig. 5c, f, i and l). Accordingly, *faf* over-expression resulted in decreased Yki activity (Fig. 6a, b and e). We also used the Yki-mediated transcriptional readout to determine whether Faf is acting through Ft to produce these effects. Over-expression of Ft in the posterior compartment of the wing resulted in a significant decrease

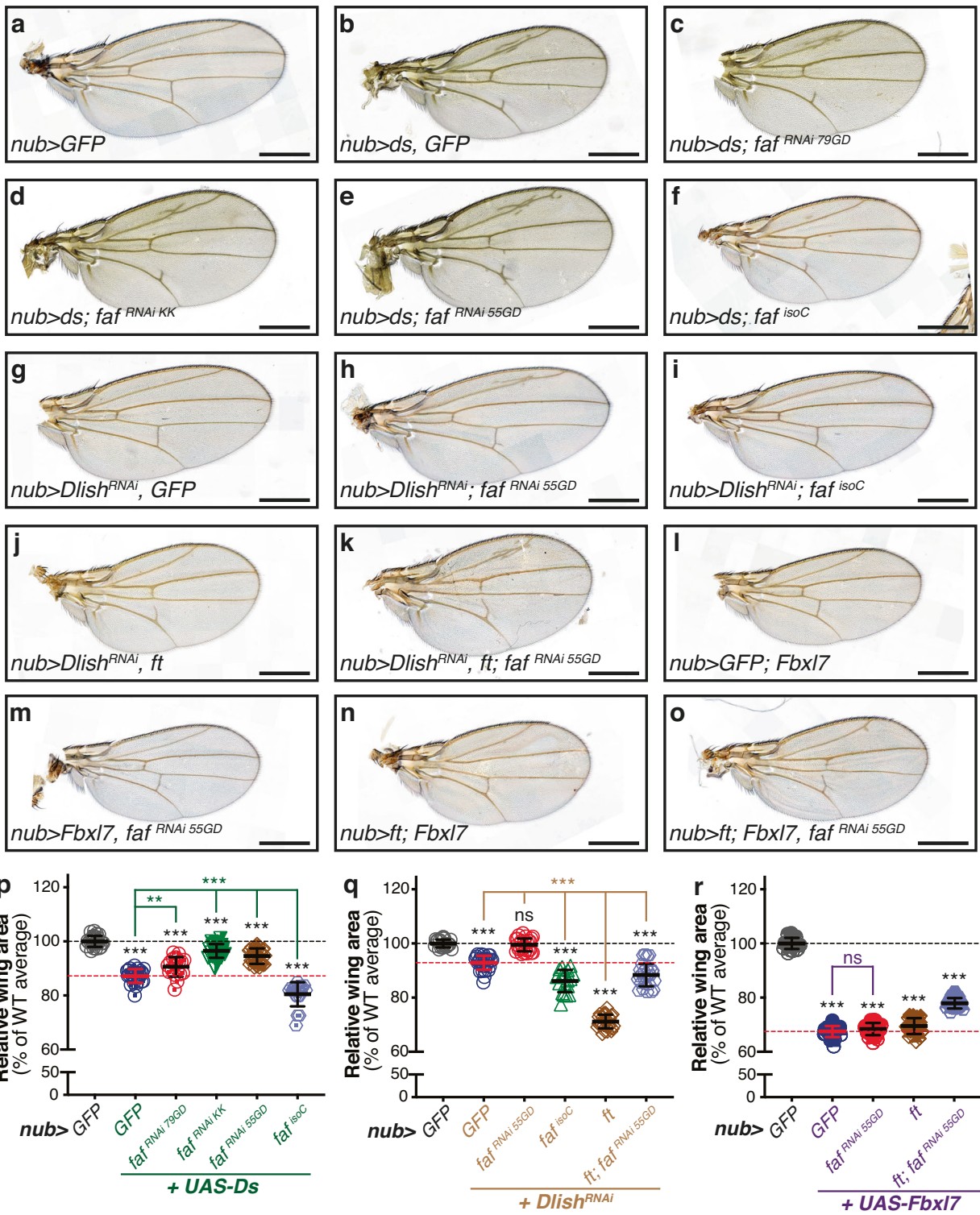

in Yki-mediated transcription, consistent with its role in activating Hippo signalling (Fig. 5d, 5f, j and l). Interestingly, depletion of *faf* completely abrogated the effect of Ft over-expression on Yki target gene expression (Fig. 5e, f, k and l). Indeed, the expression levels of the *ex-lacZ* and *DIAP1::GFP* reporters were significantly different in the *UAS-ft, UAS-faf^RNAi* condition compared to when Ft was over-expressed in isolation, and were closer to levels seen in controls. Together, this data suggests that Faf promotes tissue growth by positively regulating Hippo signalling and that it does so by modulating the function of Ft.

## The effects of Faf are dependent on its catalytic activity

Faf is part of the DUB family of proteins, enzymes that regulate ubiquitylation levels and counteract the action of E3 ubiquitin ligases in cells[50]. Given that Faf is predicted to be an active DUB, we next tested whether the role of Faf in tissue growth and modulation of Ft and Hippo signalling events was dependent on its catalytic activity. To this end, we used previously generated *UAS*-regulated transgenes encoding either WT or a catalytically inactive form of Faf (respectively, Faf^WT and Faf^CD)[36] inserted at the same genomic location. First, we assessed the role of the catalytic activity of Faf in the regulation of Yki-

**Fig. 3 | *faf* genetically interacts with *ds*, *Dlish* and *Fbxl7* in the regulation of wing tissue growth. a–o** Modulation of Faf expression levels affects the tissue growth phenotypes of *ds*- (**b–f**), *Dlish*^RNAi- (**g–k**) and *Fbxl7*-expressing flies (**l–o**). Shown are adult wings from flies raised at 25°C expressing the indicated transgenes in the wing pouch under the control of *nub-Gal4* (*nub >* ). Compared to control adult wings (**a**), wings expressing *ds* displayed significant tissue undergrowth (**b**), which was partially rescued by simultaneous depletion of *faf* (**c**, **d** and **e**) and enhanced by *faf* over-expression (**f**). RNAi-mediated depletion of *Dlish* (**g**) caused a mild reduction in wing size, which was partially suppressed by co-depletion of *faf* (**h**), and enhanced by *faf* over-expression (**i**). Expression of *ft* resulted in an enhancement of the undergrowth of *Dlish*^RNAi wings (**j**), and this was in part prevented by *faf*^RNAi (**k**). *Fbxl7* over-expression caused reduced wing size compared to controls (**l**) and this was not significantly affected by *faf*^RNAi (**m**) or *ft* expression (**n**), but this was partially

suppressed when *ft* was combined with *faf*^RNAi (**o**). **p–r** Quantification of relative adult wing sizes in genetic interactions with *ds* (**p**), *Dlish* (**q**) or *Fbxl7* (**r**). Data are represented as % of the average wing area of control wings (*nub > GFP*, which were set as 100%). Data are shown as average ± standard deviation, with all data points represented. ($n = 23, 25, 18, 30, 30$ and $18$ for (**p**); $n = 20, 28, 26, 19, 29$ and $28$ for (**q**); and $n = 23, 28, 30, 25$ and $27$ for (**r**). Black dashed lines represent average size of controls (100%) whilst red dashed lines indicate average size of adult wings from flies expressing *ds*, *Dlish*^RNAi or *Fbxl7* under the control of *nub-Gal4*. Significance was assessed using a one-way ANOVA comparing all genotypes to their respective controls (*nub > GFP*, *nub > GFP + UAS-Ds*, *nub > GFP+Dlish*^RNAi or *nub > GFP + UAS-Fbxl7*; black, green, brown or blue asterisks, respectively), with Dunnett's multiple comparisons test. **, $p < 0.01$; ***, $p < 0.001$. n.s. non-significant. Scale bar: 500 μm.

dependent transcription using *ex-lacZ* (Fig. 6a–e) and *HRE-DIAP1::GFP* (Figure S9a–e). Over-expression of Faf resulted in a reduction in *ex-lacZ* levels when compared to the *UAS-GFP* control (Fig. 6a–c and e). Similar results were obtained when the *DIAP1::GFP* reporter was analysed (Figures S9a–c and S9e). These results confirm the effect of Faf on tissue growth and support the hypothesis that, at least in part, the role of Faf involves the regulation of Hippo signalling activity. Interestingly, the catalytic mutant version of Faf (*faf*^CD) had no effect on either the *ex-lacZ* (Fig. 6d and e) or the *DIAP1::GFP* reporters (Figure S9d and S9e). This strongly suggests that the role of Faf in the regulation of tissue growth and Hippo signalling is dependent on its catalytic activity and DUB function, as opposed to a potential role as a scaffolding protein bridging protein-protein interactions.

To further validate these observations, we assessed other Ft-related phenotypes, such as the regulation of D subcellular localisation. As shown in Figures S8 and S9, Faf over-expression affected D membrane localisation and increased D cytoplasmic levels. These results were recapitulated when we overexpressed *faf*^WT in the developing wing disc (Figure S9g). Compared to the respective anterior compartment and controls (*lacZ*^RNAi), tissues where *faf*^WT was overexpressed had D subcellular localisation defects. In contrast, when we overexpressed the *faf*^CD mutant, we observed no overt effects on the levels or subcellular localisation of D (Figure S9h). As before, no changes in Arm localisation were observed (Figure S9i–k). Again, this suggests that the effect of Faf on Ft signalling events is dependent on its catalytic activity.

We also directly assessed the role of Faf DUB activity in the regulation of tissue growth and on the genetic interactions with Ft. For this, we expressed Faf^IsoC, Faf^WT and Faf^CD in the wing pouch using *nub-Gal4* and measured wing size as a proxy for the effects of Faf on tissue growth. When tested in isolation, as expected, we observed phenotypes consistent with the proposed role of Faf in the regulation of tissue growth (Fig. 6f-j). Adult wings from animals expressing *UAS-faf*^IsoC (Fig. 6g) or *UAS-faf*^WT (Fig. 6h) were significantly smaller than those of the controls (*nub > GFP*, Fig. 6f and j). Interestingly, the phenotype elicited by *UAS-faf*^WT was more severe than *UAS-faf*, which can be explained by the fact that the transgenes have different genomic locations and, therefore, different expression levels. Indeed, RT-PCR experiments revealed increased expression of *faf* in larvae expressing *faf*^IsoC and this was enhanced in *faf*^WT-expressing flies (Figure S9l). Importantly, when *UAS-faf*^CD was expressed, we did not observe tissue growth restriction but, instead, the adult wings were slightly larger than controls (Fig. 6i). This suggests that the catalytic activity of Faf is required for its effect on tissue growth. Moreover, the mild overgrowth of Faf^CD wings raises the possibility that this catalytically inactive allele may be acting as a mild dominant negative version of Faf.

Next, we assessed genetic interactions between Faf and Ft by expressing the Faf transgenes in the presence of Ft over-expression (Fig. 6k-o) or Ft RNAi-mediated depletion (Fig. 6p-t). As previously shown, depletion of *ft* resulted in overgrowth phenotypes (Fig. 6p). In agreement with our previous results, expression of *UAS-faf*^IsoC in these

conditions abrogated this overgrowth phenotype (Fig. 6q). Similarly, expression of *UAS-faf*^WT blocked the overgrowth caused by depletion of *ft* and, in fact, caused a significant undergrowth phenotype (Fig. 6r). Again, in agreement with our hypothesis, expression of the inactive form of Faf (*UAS-faf*^CD) did not modify the phenotype of *ft*^RNAi flies and the adult wing sizes were indistinguishable from those of controls (*nub > ft*^RNAi, *GFP*) (Fig. 6s and t). We validated these observations by assessing the effect of co-expression of Ft and Faf (Fig. 6k-o). Ft over-expression caused a significant undergrowth phenotype when compared to controls (Fig. 6k and f), which was enhanced by co-expression of either *UAS-faf*^IsoC (Fig. 6l) or *UAS-faf*^WT (Fig. 6m). In contrast, expression of the mutant *UAS-faf*^CD resulted in a mild rescue of the *UAS-ft* adult wing phenotype (Fig. 6n and o). Together, our in vivo data support our conclusion that the catalytic activity of Faf is essential for its function in the regulation of Hippo signalling and tissue growth.

## Faf-mediated regulation of Ft protein is dependent on its DUB activity

Our in vivo results strongly suggest that Faf DUB activity is required for its function in tissue growth. To test if this effect was directly connected to Ft regulation, we assessed the effect of WT and mutant Faf on the protein levels of Ft, in vivo and in vitro. Firstly, we used *hh-Gal4* to express the *faf* transgenes in the posterior compartment of the developing wing imaginal disc and monitored Ft protein levels using a Ft-specific antibody (Fig. 7a–d). We observed that, in agreement with our previous observations, expression of *faf*^WT resulted in an increase in Ft protein levels (expressed as the ratio between the levels of Ft in the posterior and anterior compartments or normalised to the corresponding Arm levels) (Fig. 7b, d and f), whereas the expression of the *faf*^CD catalytic mutant had no obvious effect on Ft antibody staining (Fig. 7c, d and f). Importantly, despite the fact that *faf*^WT caused significant changes in wing disc morphology, none of the Faf transgenes affected the protein levels of Arm, which was used as a control for a global effect on apical membrane proteins (Fig. 7e).

Next, we sought to confirm these observations in vitro in *Drosophila* S2 cells. For that, we generated a catalytically inactive version of Faf (Faf^CD) in our Faf^LD cDNA clone and tested whether its expression modulated Ft protein levels. In contrast to WT Faf, which promoted the stabilisation of Ft and resulted in higher levels of Ft protein in cell lysates, expression of Faf^CD in S2 cells had a minimal effect on Ft levels (Fig. 7g and h). Importantly, as in this situation we assessed the levels of epitope-tagged Ft rather than its endogenous levels, the effects of Faf are likely due to post-translational modifications and not via the regulation of endogenous *ft* gene transcription and/or translation. Accordingly, we observed that treating S2 cells with the proteasome inhibitor MG132 resulted in higher Ft protein levels (Figure S10a and S10b). Notably, Faf expression stabilised Ft protein levels to such extent that the effect of MG132 was minimal (Figure S10a). As previously observed, depletion of *faf* resulted in destabilisation of Ft, which was partly rescued by MG132 treatment, suggesting that Ft is degraded by the proteasome when Faf is absent (Figure S10b). Given

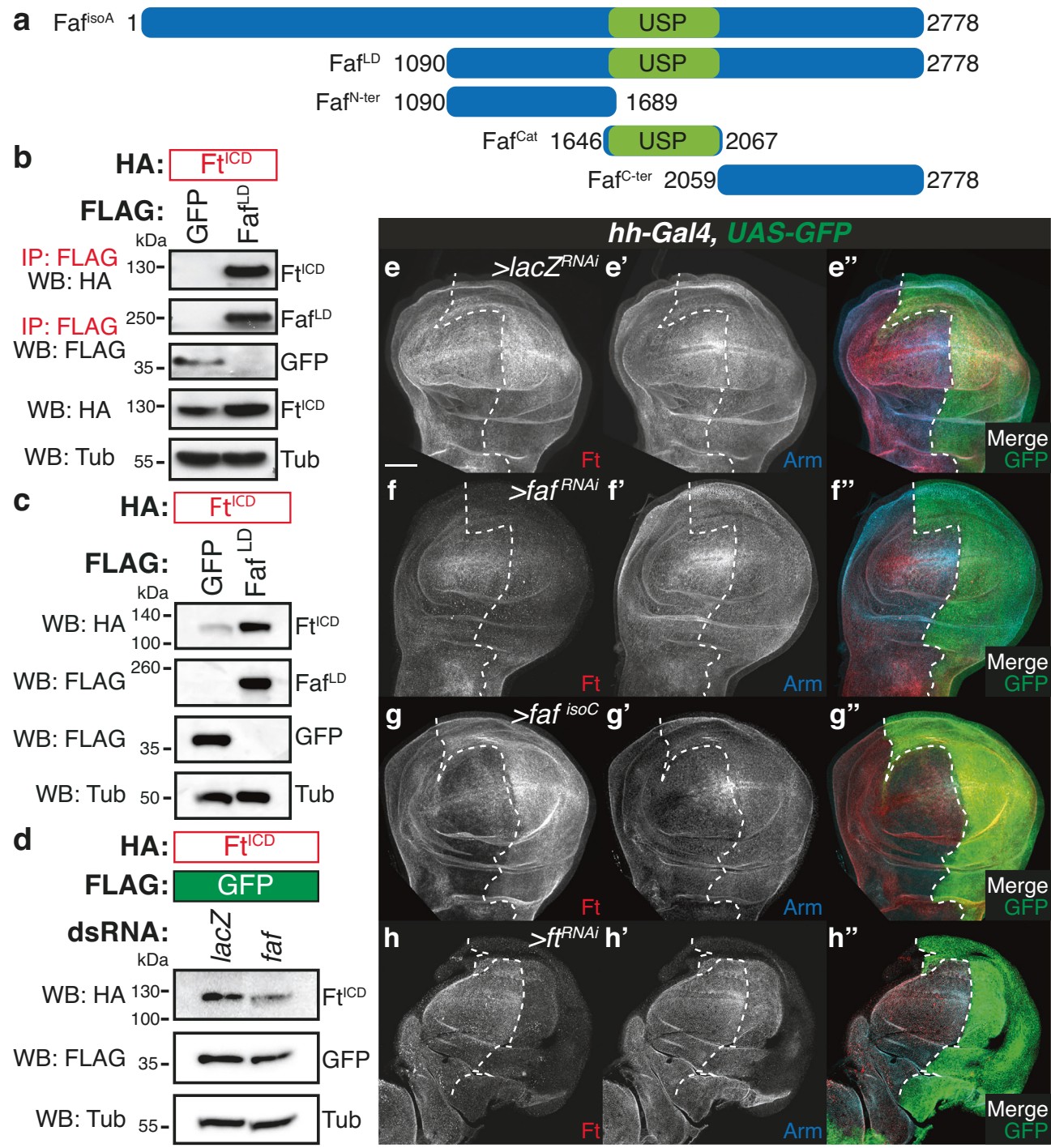

**Fig. 4 | Faf interacts with Ft and regulates its protein levels in vitro and in vivo. a** Schematic representation of the Faf constructs used in this study. Numbers denote amino acid position. USP DUB domain is represented in green. **b** Faf binds to Ft. HA-tagged Ft$^{ICD}$ was co-expressed with FLAG-tagged GFP or Faf$^{LD}$ in *Drosophila* S2 cells. Cells were lysed and lysates were subjected to co-immunoprecipitation using FLAG agarose beads. Lysates were analysed by immunoblot using the indicated antibodies for detection of protein expression and co-purification. Tubulin (Tub) was used as loading control. (*n* = 3 independent experiments). **c, d** Faf regulates Ft protein levels in *Drosophila* S2 cells. **c** HA-tagged Ft$^{ICD}$ was expressed in S2 cells in the presence or absence of Faf$^{LD}$. 48 h after transfection, cells were lysed and lysates were analysed by immunoblot using the indicated antibodies. Ø represents expression of empty vector. **d** S2 cells were treated with the indicated dsRNAs 24 h before co-transfection with the indicated constructs. Ft$^{ICD}$ protein levels were

analysed by Western blotting with the indicated antibodies 48 h after cell transfection. GFP and Tubulin (Tub) were used as transfection and loading control, respectively. (*n* = 3 independent experiments). **e–h** Faf regulates Ft protein levels in vivo. Shown are XY confocal micrographs of third instar wing imaginal discs expressing the indicated constructs under the control of *hh-Gal4*, showing Ft antibody staining (e-h; red in e''-h''), Arm antibody staining (e'-h'; cyan in e''-h''), and direct fluorescence from GFP (green in e''-h''). Compared to the controls (**e**, *lacZ$^{RNAi}$*), *faf* depletion (**f**) and *faf* over-expression (**g**) resulted in a decrease or increase in Ft protein levels, respectively. Shown is also *ft$^{RNAi}$* control (**h**) to validate antibody specificity. Ventral is up in XY sections, whilst GFP marks the *hh-Gal4*-expressing posterior compartment (right). Dashed white line depicts boundary between anterior and posterior compartments. (*n* = 11, 14, 12, 8). Scale bar: 50 μm.

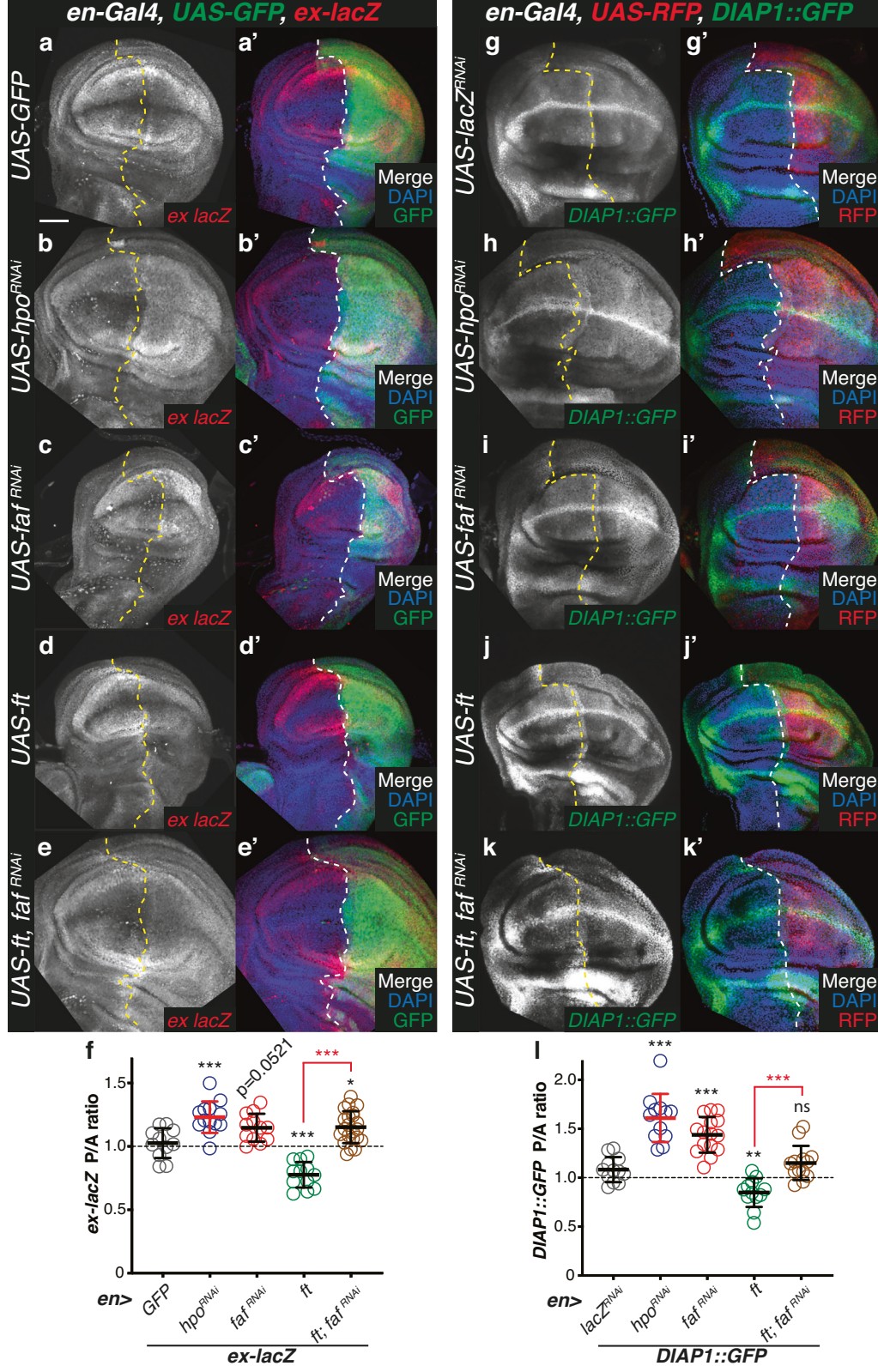

the prominent effects of modulating Faf expression on Ft protein levels and the limited effect of MG132, it is possible that Ft protein degradation involves other mechanisms beyond proteasome-mediated degradation. Our data is consistent with the notion that Faf regulates Ft in a DUB-dependent manner. We further tested this possibility by treating *Drosophila* S2 cells with a DUB inhibitor that affects Faf function, WP1130[51,52] (Fig. 7i and j). WP1130-treated cells

exhibited lower levels of Ft protein than vehicle-treated cells (Fig. 7i and j). Although WP1130 is thought to inhibit other DUBs[51,52], the effect on Ft levels in S2 cells appears to be due to its effect on Faf since Faf over-expression largely abrogated the effect of WP1130 and *faf* depletion resulted in reduced Ft levels that were only minimally affected by WP1130 treatment, suggesting that other targets for this inhibitor do not play a major role (Figure S10c). Together, our results

**Fig. 5 | Faf regulates expression of Yki target genes in vivo. a–e** Regulation of *ex-lacZ* expression. XY confocal sections of third instar wing imaginal discs containing *ex-lacZ* (a-e; red in a'-e' merged images), in which *en-Gal4* was used to drive expression of *UAS-GFP* (**a**), *UAS-hpo^RNAi^* (**b**), *UAS-faf^RNAi^* (**c**), *UAS-ft* (**d**) or *UAS-ft* and *UAS-faf^RNAi^* (**e**). GFP (green in a'-e' merged images) indicates posterior compartment where transgenes are expressed. DAPI (blue) stains nuclei. Dashed lines indicate anterior-posterior compartment boundary. **f** Quantification of *ex-lacZ* expression levels. Shown are the posterior/anterior (P/A) *ex-lacZ* ratios for the different genotypes analysed. Data are shown as average ± standard deviation, with all data points represented. (*n* = 11, 13, 12, 12 and 22). **g-k** Regulation of *DIAP1::GFP* expression. XY confocal sections of third instar wing imaginal discs carrying *DIAP1::GFP* (g-k, shown in green in g'-k' merged images), in which *en-Gal4* was used to drive expression of *UAS-lacZ^RNAi^* (**g**), *UAS-hpo^RNAi^* (**h**), *UAS-faf^RNAi^* (**i**), *UAS-ft* (**j**) or *UAS-ft* and *UAS-faf^RNAi^* (**k**). RFP (red in g'-k' merged images) indicates posterior compartment where transgenes are expressed. DAPI (blue) stains nuclei. Dashed lines indicate boundary between anterior and posterior compartments. **l** Quantification of *DIAP1::GFP* expression levels. Shown are posterior/anterior (P/A) *DIAP1::GFP* ratios for the indicated genotypes. Data are shown as average ± standard deviation, with all data points represented. (*n* = 12, 12, 15, 12 and 13). Significance was assessed using a one-way ANOVA comparing all genotypes to controls (*UAS-GFP* in (**f**) and *UAS-lacZ^RNAi^* in (**l**)), with Dunnett's multiple comparisons test. Pairwise comparisons between *UAS-ft* and *UAS-ft; UAS-faf^RNAi^* (red) were performed using unpaired two-tailed t-test with Welch's correction. *, $p < 0.05$; **, $p < 0.01$; ***, $p < 0.001$; ns, non-significant. Scale bar: 50 μm.

reinforce the notion that the catalytic activity of Faf is important for its modulation of Ft protein levels and, subsequently, of Ft-mediated signalling.

## Evolutionary conservation of Faf function in tissue growth regulation

Given that Faf is part of an evolutionarily conserved protein family with recognisable orthologues in other species, we next sought to determine whether the function of Faf in the regulation of Ft-mediated signalling is conserved. For this, we initially tested the effect of the mammalian orthologue of Faf, USP9X, in vivo in *Drosophila* tissues. Using the *en-Gal4* driver, we expressed *UAS-USP9X* in the posterior compartment of the developing wing and assessed whether this impacted on the expression of the reporters of Yki activity, *ex-lacZ* or *HRE-DIAP1::GFP* (Fig. 8a–f). Like *UAS-faf^WT^*, expression of its mammalian paralog *USP9X* resulted in a reduction in the posterior/anterior (P/A) ratio of *ex-lacZ* and *DIAP1::GFP* expression (Fig. 8a–f). We also determined whether USP9X disrupted D subcellular localisation in the wing disc. Expression of *USP9X* resulted in a disordered D localisation in the wing epithelial cells (Figure S10d and S10e). Similarly to *UAS-faf* (Figure S8e' and S8e''), we observed that cells expressing *UAS-USP9X* in the posterior compartment, in several instances, appeared to display D discontinuously at the apical membrane or to lack it altogether (Figure S10e'). Importantly, Arm localisation was unaffected (Figure S10f and S10g). These data suggest that, at least when overexpressed, USP9X phenocopies Faf in *Drosophila* tissues, supporting the hypothesis that the function of Faf in tissue growth is conserved in mammalian tissues.

To test this idea further, we analysed tissue growth parameters in *Drosophila* adult wings using *nub-Gal4* to express *UAS-USP9X*. In agreement with the effect of USP9X on Yki-mediated gene expression, expression of *UAS-USP9X* in the wing pouch resulted in an undergrowth phenotype (Fig. 8h), when compared with controls (Fig. 8g). Tissue growth was significantly reduced, as evidenced in Fig. 8i. We also tested whether USP9X was able to genetically interact with Ft by combining *UAS-USP9X* with Ft over-expression or RNAi-mediated depletion. Co-expression of *ft* and *USP9X* enhanced the undergrowth phenotype of *UAS-ft* wings (Fig. 8j–l), indicating that Ft downstream signalling is potentially more active in the presence of ectopic USP9X. In contrast to *ft* over-expression, depletion of *ft* (*UAS-ft^RNAi^*) resulted in enhanced tissue growth in the wing (Fig. 8m). Expression of *USP9X* in these conditions resulted in a suppression of the *UAS-ft^RNAi^* phenotype (Fig. 8n), and a return to WT wing tissue size (Fig. 8o). This is consistent with the proposed effect of Faf on the regulation of Ft protein levels and suggests that USP9X retains at least some of the functions of Faf in this context.

## USP9X-mediated regulation of Ft is conserved

Next, we assessed whether the effects seen with USP9X over-expression in *Drosophila* tissues are related to its potential regulation of Ft protein levels. To test this, we first monitored Ft protein levels in the *Drosophila* wing imaginal disc using *hh-Gal4* to control

USP9X expression (Fig. 9a–e). Compared with the respective control (*hh > lacZ^RNAi^*; Fig. 9a), Ft protein levels were increased in the posterior compartment of *hh > USP9X* wing imaginal discs (Fig. 9b, c and e), while Arm levels were unaffected (Fig. 9a', b' and d).

We next tested whether USP9X could control Ft protein levels in mammalian cells. In mammals, there are multiple genes encoding Ft cadherins, *Fat1-4*[12]. However, taking into account protein homology and function, Fat4 is the closest mammalian orthologue of Ft[12]. Therefore, we assessed if the stabilisation of Ft by Faf could be recapitulated in a mammalian setting using the corresponding mammalian proteins, USP9X and Fat4. To this end, we used plasmids encoding different Fat4 truncations (Fat4^ICD^ or Fat4^ΔECD^), alongside a plasmid encoding USP9X and expressed them in HEK293 cells. Cells were transfected with Fat4 alone or in combination with USP9X and Western blot analysis of cell lysates revealed that Fat4 levels were strongly increased in the presence of USP9X (Figs. 9f, g and S10h). Importantly, this effect was observed for both Fat4^ICD^ and Fat4^ΔECD^ (Fig. 9f and S10h). Given that both Fat4^ICD^ and Fat4^ΔECD^ largely lack the extracellular domain of Fat4, our data indicates that the effect of USP9X on the levels of Fat4 is likely to be independent of any interactions with its cognate partners that associate with the Fat4 extracellular domain. Moreover, since we assessed a FLAG-tagged Fat4 version rather than the endogenous Fat4, the USP9X-mediated regulation of Fat4 levels is predicted to be primarily the result of a post-translational mechanism, rather than an indirect effect on *Fat4* gene expression.

We sought to validate our observations by directly assessing Fat4 levels in immunofluorescence experiments (Fig. 9h–j). For this, HEK293 cells were transfected with GFP and either Fat4 (Figure S10i) or USP9X (Fig. 9h and i) and stained with an anti-Fat4 antibody. Subsequently, Fat4 protein levels were analysed and compared between control GFP-negative cells and USP9X-expressing, GFP-positive cells. Analysis of our immunofluorescence experiments revealed that cells expressing GFP and USP9X exhibited higher levels of Fat4 than cells that did not express GFP (Fig. 9h–j). Taken together, our data indicates that, similarly to the role of Faf in *Drosophila*, USP9X stabilises Fat4 protein in mammalian cells.

## Discussion

Ft is an atypical cadherin with essential functions in tissue growth and cell polarity[12,53]. Despite intense study into its cellular role, the mechanisms regulating Ft function and its downstream effects remain relatively elusive. Ft interacts, both genetically and physically, with many proteins involved in tissue growth and planar cell polarity regulation[11,53,54], but it is still unclear how these two functions and the multiple interactions are controlled and coordinated. Interestingly, several Ft-associated proteins and processes are thought to be linked to protein ubiquitylation (e.g., Dlish[26], D[17,20], Fbxl7[30,31] and Elgi[32]). Surprisingly, despite these previous reports, the action of DUBs has not been associated with the regulation of *Drosophila* Ft. Here, we identified Fat facets (*faf*) as a regulator of Ft and delineated its role in the regulation of Ft protein stability and function.

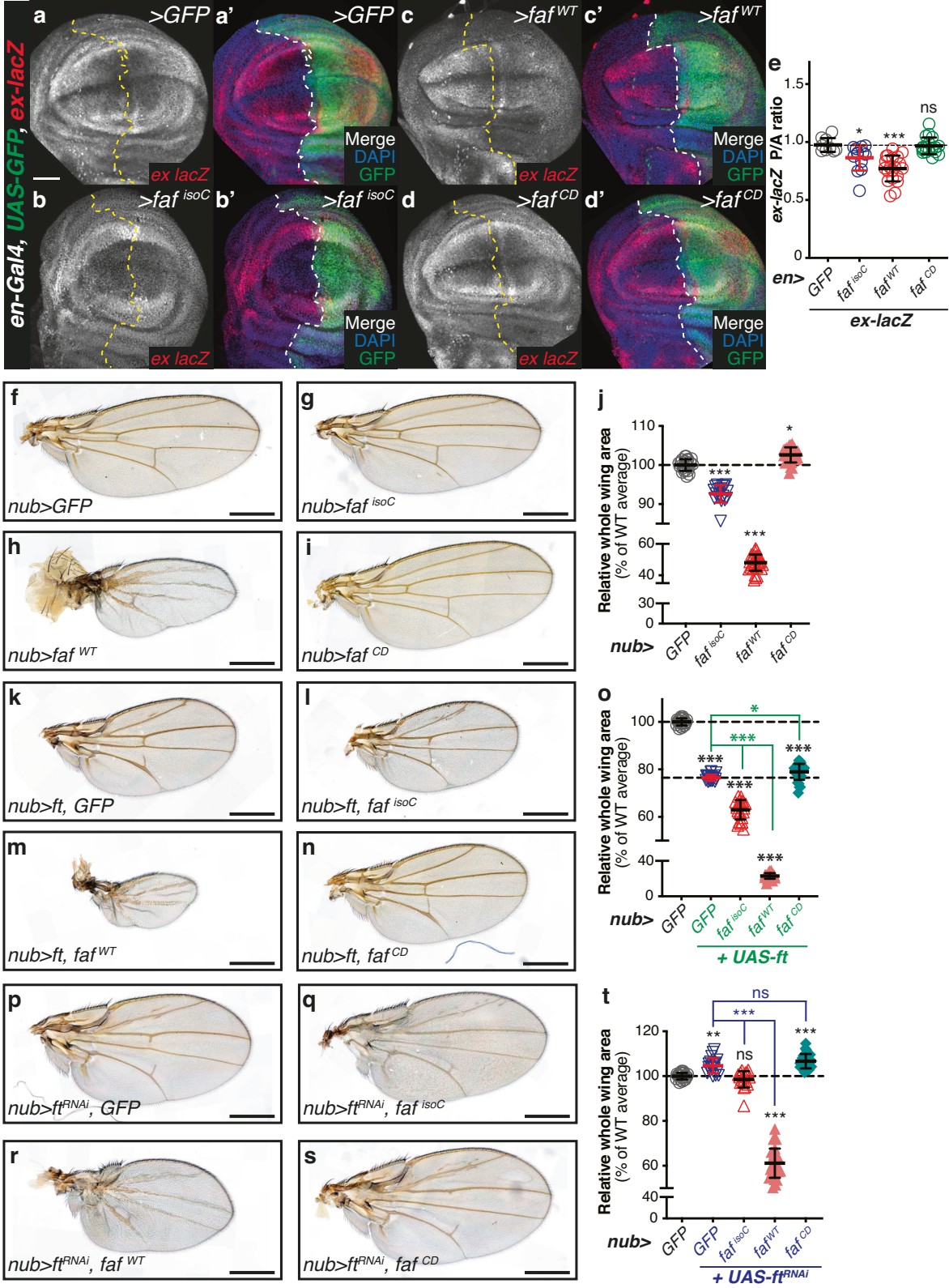

Our observations strongly suggest that Faf modulates the role of Ft in the regulation of tissue growth and Hippo signalling, in a manner consistent with a mechanism involving the regulation of Ft protein levels. Tissue growth phenotypes associated with increased or decreased levels of Ft in developing tissues were abrogated or enhanced when Faf levels were modified by over-expression or RNAi-mediated depletion, in agreement with a role for Faf in promoting Ft

protein stability. In addition, Faf affected Hippo signalling readouts in vivo. In fact, the effect of Ft on Hippo signalling appears to be partly dependent on Faf function. We also observed a genetic interaction between *faf* and *ft*, which further suggests that Faf function is important for the role of Ft in the regulation of tissue growth. However, we cannot fully exclude the hypothesis that Faf also has a parallel function that impinges on imaginal disc growth, particularly as Faf has been

**Fig. 6 | The role of Faf in modulating Ft function is dependent on its DUB activity. a–d** Faf regulates Yki target gene expression in a DUB-dependent manner. XY confocal sections of third instar wing imaginal discs carrying *ex-lacZ* (**a–d**, shown in red in **a'–d'** merged images), in which *en-Gal4* was used to drive expression of *GFP* (**a**), *faf^isoC* (**b**), *faf^WT* (**c**) or *faf^CD* (**d**). GFP (green in a'-d' merged images) indicates posterior compartment where transgenes are expressed. DAPI (blue) stains nuclei. Dashed lines indicate anterior-posterior compartment boundary. **e** Quantification of *ex-lacZ* expression levels. Shown are the posterior/anterior (P/A) *ex-lacZ* ratios for the different genotypes analysed. Data are shown as average ± standard deviation, with all data points represented. (*n* = 9, 13, 20 and 18). Significance was assessed using a one-way ANOVA comparing all genotypes to *UAS-GFP*, with Dunnett's multiple comparisons test. *, $p < 0.05$; ***, $p < 0.001$; ns, non-significant. Scale bar: 50 μm. **f–t** Effect of Faf on Ft-mediated regulation of growth is DUB-dependent. Shown are adult wings from flies raised at 25°C expressing transgenes under *nub-Gal4* control. (**j**), (**o**), (**t**) Quantification of effect of Faf on tissue growth. Shown are relative adult wing sizes from flies expressing the indicated transgenes under *nub-Gal4* control. Data are represented as % of the average wing area of controls (*nub > GFP*, set to 100%). Data are shown as average ± standard deviation, with all data points depicted. (*n* = 22, 18, 30, 26 for j; *n* = 22, 21, 22, 29, 32 for o; and *n* = 22, 19, 19, 29 and 25 for t). Significance was assessed using a one-way ANOVA analysis comparing all genotypes to *nub > GFP* (black asterisks), to *nub > UAS-ft* (green asterisks), or to *nub > UAS-ft^RNAi* (blue asterisks) with Dunnett's multiple comparisons test. *, $p < 0.05$; **, $p < 0.01$; ***, $p < 0.001$; ns, non-significant. Scale bar: 500 μm.

previously associated with the Notch, RTK/Ras and JNK pathways[33,34,36,55–57].

Our results suggest that Faf mainly regulates the tissue growth function of Ft, rather than its PCP functions, since Faf affected Ft-induced tissue size, but not tissue shape phenotypes. This observation was somewhat unexpected, given that Faf has been previously shown to regulate the protein levels of core PCP proteins in the wing via the regulation of Flamingo (Fmi)[58] and that its mammalian orthologue USP9X has been associated with PCP regulation via the deubiquitylation of DVL2[59]. However, we have not directly addressed whether Faf regulates Ft-mediated PCP downstream readouts and, therefore, cannot rule out a function for Faf in modulating Ft-related PCP functions. Nevertheless, our results suggest that a putative function of Faf in regulating PCP in the context of Ft signalling is not as relevant as its regulation of tissue growth. It is also possible that ubiquitylation may be a signal or a switch controlling the role of Ft and directing it toward tissue growth regulation or modulation of PCP, by altering Ft protein-protein interactions. Further investigation is required to reconcile these observations. Alternatively, or in addition, Ft ubiquitylation may associated with establishing a specific threshold of Ft signalling at particular developmental stages or in specific tissues, which then influence the response to specific signals, such as mechanical forces.

The effects of Faf on tissue growth and Hippo signalling regulation in *Drosophila* were recapitulated with the mammalian orthologue USP9X, suggesting that the role of Faf in regulating Ft function may be evolutionarily conserved. Indeed, USP9X has been previously associated with Hippo signalling regulation, albeit not with the regulation of the Ft orthologue Fat4. USP9X is thought to regulate Hippo signalling via the modulation of YAP1[60], LATS[61,62] and Angiomotin proteins[63,64]. In addition, USP9X has been shown to interact with and to deubiquitylate YAP1[60], LATS[61,62], WW45[61], KIBRA[61], AMOT[61,64] and AMOTL2[63]. It is still unclear how USP9X affects Hippo signalling, as some of its reported substrates have opposing effects on the pathway. Whilst some of the differences can be potentially explained by cell-type-specific functions of USP9X, these are not sufficient to reconcile all previous observations. Moreover, none of these reports has assessed the role of USP9X in regulating Fat4 or other mammalian Ft proteins. Therefore, it remains a possibility that some of the previously reported effects of USP9X on Hippo signalling could be related to Fat4 regulation, particularly for those proteins known to be in the vicinity of Fat4 at the membrane. Interestingly, in the mouse heart, Fat4 regulates Hippo signalling and YAP1 activity via the Angiomotin protein Amotl1[65], raising the possibility that USP9X may be required for this function. USP9X has a complex role in cancer, with both pro- and anti-tumourigenic functions and this may be related to the fact that it not only regulates Hippo signalling, but also other pathways such as TGF-β, Wnt and JAK-STAT, both in mammals[59,66–70], as well as in *Drosophila*[35].

Our data firmly establishes Faf as a regulator of Ft function and tissue growth. However, questions remain regarding the precise molecular mechanisms involved. Our data is consistent with a direct effect of Faf on Ft, given that the two proteins interact in co-IP experiments. Faf controls Ft protein levels post-translationally in a catalytically-dependent manner, which is supported by the effect of the DUB inhibitor WP1130 on Ft protein levels. Although WP1130 is thought to inhibit other DUBs (USP5, USP14, UCH37 and UCH-L1)[52], the orthologous *Drosophila* genes were either not identified as hits in our in vivo RNAi screen or had the opposite effect to Faf, suggesting that Faf is the relevant target for WP1130 in the context of Ft regulation.

However, whilst the DUB activity of Faf seems to be required for the regulation of Ft protein levels and function, it is still unclear whether Faf directly deubiquitylates Ft or if its effect is indirect. Analysis of a direct effect of Faf in deubiquitylating Ft is complicated by the lack of evidence regarding direct Ft ubiquitylation and the fact that no E3 ligase has been identified that targets Ft. Previous efforts have failed to establish Elgi as a Ft E3 ligase[32] and, therefore, we have not tested whether Faf-mediated regulation of Ft is related to the E3 ligase activity of Elgi. Without a clear E3 ligase candidate to promote ectopic Ft ubiquitylation in *Drosophila* S2 cells, it would be challenging to firmly establish Faf as a direct Ft DUB. Alternative models that are not dependent on Faf directly acting on Ft are also plausible. For instance, although the precise mechanistic details remain unclear, in their role as modulators of Ft-mediated signalling, both Fbxl7 and Elgi have been proposed to regulate protein trafficking[30,32]. Therefore, it is possible that the effect of Faf is also related to protein trafficking. Indeed, it has been previously reported that the mechanism by which Faf regulates PCP in *Drosophila* is potentially related to protein trafficking, as loss of *faf* results in increased lysosomal degradation of Fmi, a possible consequence of improper vesicle accumulation of internalised Fmi[58]. Moreover, Faf and its putative substrate Lqf (Liquid facets, a *Drosophila* Epsin) have been proposed to control endocytosis of several cargoes, including the Notch ligand, Delta[55,57]. Whether Faf-mediated regulation of Ft is dependent on Ft endocytosis and trafficking is still unknown.

Finally, while the level at which Faf acts within the Ft signalling pathway remains to be precisely determined, some hints come from our in vivo genetic interaction experiments, the analysis of D subcellular localisation, and the effects on Ft protein stability. Our data suggest that Faf likely acts at a regulatory node including Ft, Fbxl7 and App and, given the phenotypes observed, it is possible that it antagonises the action of Dlish in regulating D function. This is supported by the fact that Faf appears to act upstream of the Hpo kinases Hpo and Wts. Further biochemical experiments are required to determine if the interactions between these Ft signalling components are modulated by Faf, if App-mediated palmitoylation is affected by Faf activity, or if any of these proteins is a bona fide Faf substrate. Another question to address in future experiments is whether Faf function is regulated by specific cues, for instance mechanical forces, nutrient supply and hormonal signals, all inputs that control Hpo signalling and, by extension, tissue growth.

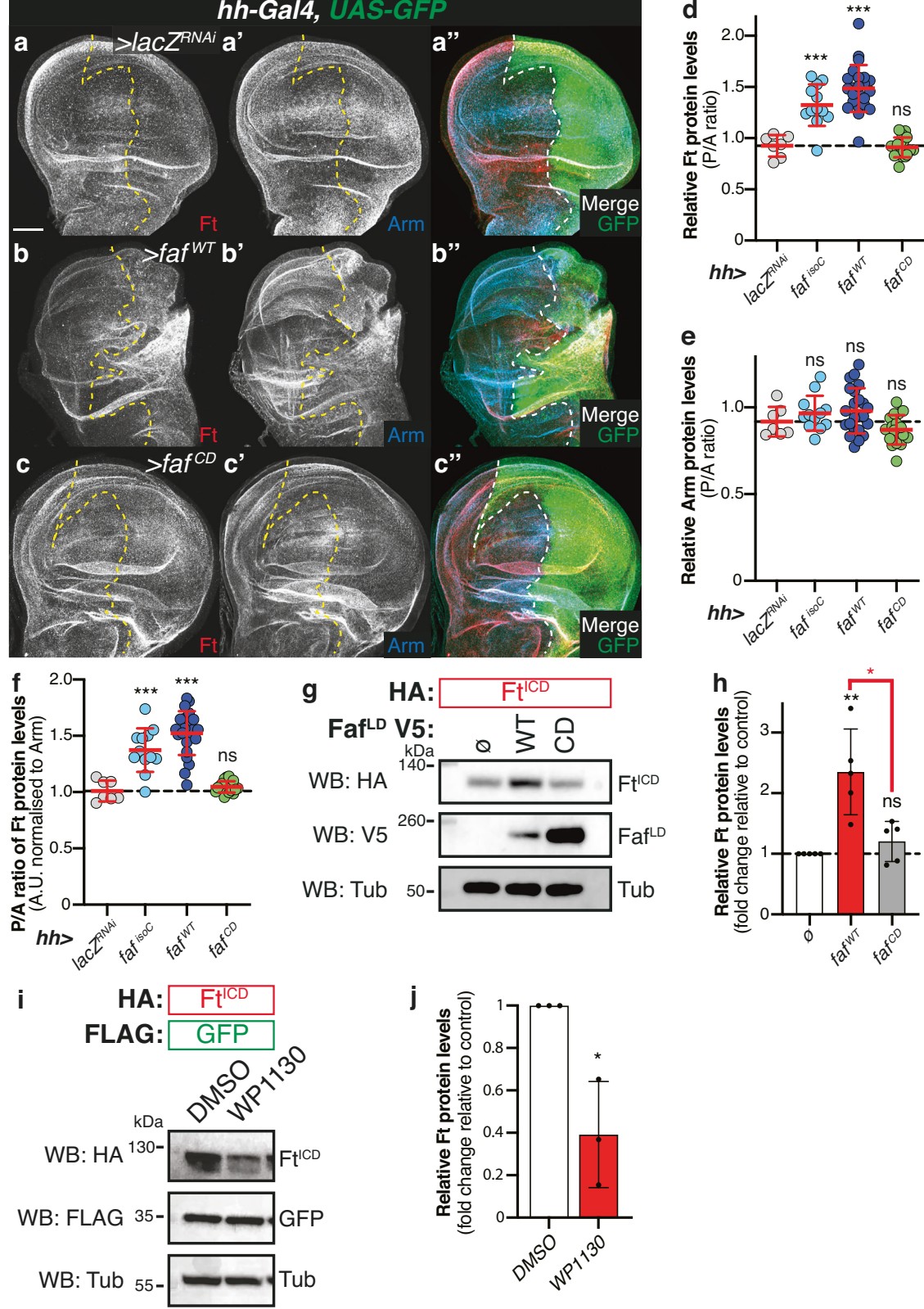

## Methods

### Drosophila cell culture, expression constructs and chemical treatments

Work involved the use of the *Drosophila* cell line Schneider S2 (RRID:CVCL_Z232), obtained from the ATCC and screened for mycoplasma presence, showing no contamination. *Drosophila* S2 cells were grown in *Drosophila* Schneider's medium (Thermo Fisher Scientific)

supplemented with 10% (v/v) FBS, 50 μg/mL penicillin and 50 μg/mL streptomycin. Expression plasmids were transfected using Effectene transfection reagent (QIAGEN). Plasmids were generated via Gateway® technology (Thermo Fisher Scientific). Open reading frames (ORFs) were PCR amplified from cDNA clones obtained from the *Drosophila* Genomics Resource Centre (DGRC, https://dgrc.cgb.indiana.edu/vectors/Overview) and cloned into the pDONR207 or pDONR-Zeo

**Fig. 7 | Faf regulation of Ft protein levels is DUB-dependent. a–c** Faf regulates Ft protein levels in vivo in a DUB-dependent manner. XY confocal micrographs of third instar wing imaginal discs expressing the indicated constructs under the control of *hh-Gal4*, showing Ft (**a–c**; red in **a″-c″**) or Arm antibody staining (**a′-c′**; cyan in **a″-c″**), and direct GFP fluorescence (green). Ventral is up in XY sections, whilst GFP marks the *hh-Gal4* compartment (right). Dashed white line depicts anterior-posterior compartment boundary. Scale bar: 50 μm. **d–f** Quantification of in vivo Arm and Ft protein levels. Shown are posterior/anterior (P/A) ratios for Fat (**d**), Arm protein levels (**e**), and normalised Ft protein levels ((**f**); normalised to Arm). Data are shown as average ± standard deviation, with all data points represented. (*n* = 7, 12, 23 and 18). Significance was assessed using a one-way ANOVA comparing all genotypes to the control (*hh>lacZ^{RNAi}*), with Dunnett's multiple comparisons test. **, *p* < 0.01; ***, *p* < 0.001. **g–j** Regulation of Ft protein levels in S2 cells requires the DUB activity of Faf. **g** HA-tagged Ft^{ICD} was co-expressed with V5-tagged Faf (WT or catalytically-inactive version, CD) in *Drosophila* S2 cells. Lysates were analysed by immunoblot using the indicated antibodies. Tub (Tub) was used as loading control. **h** Quantification of effect of Faf catalytic activity on Ft levels. Shown are relative Ft protein levels (fold change relative to controls, set to 1 in cells transfected with empty plasmid (ø); Ft levels were normalised to Tubulin) quantified from Western blot experiments. Data are represented as average ± standard deviation, with all data points represented. (*n* = 5 independent experiments). Significance was assessed by one-way ANOVA comparing all samples to controls (ø), with Tukey's multiple comparisons test or an unpaired two-tailed t-test. *, *p* < 0.05 **, *p* < 0.01. **i** DUB inhibition affects Ft protein stability. S2 cells were treated with 5 μM of WP1130 for 6 h before lysis and immunoblot analysis with the indicated antibodies. FLAG-tagged GFP and Tub were used as transfection and loading controls, respectively. **j** Quantification of effect of DUB inhibition on Ft protein levels. (*n* = 3 independent experiments). Significance was assessed by an unpaired two-tailed t-test. *, *p* < 0.05.

Entry vectors. Destination vectors used were obtained from the *Drosophila* Gateway Vector Collection or generated in-house as previously described[28]. All Entry vectors were verified by sequencing. Point mutations were generated using the Quikchange Site-Directed Mutagenesis kit (Agilent) according to the manufacturer's instructions. The Ft^{ICD} plasmid has been previously described[43]. Where indicated, inhibition of Faf was achieved by treating cells with 5 μM of the DUB inhibitor WP1130, also known as Degrasyn (Cambridge Bioscience) for 6 h before cell lysis. Proteasome inhibition was achieved by treating S2 cells with 50 μM of MG132 (Cambridge Bioscience) for 4 h before cell lysis.

## Mammalian cell culture and expression constructs

Mammalian in vitro work involved the use of HEK293 cells (RRID:CVCL_0045), obtained from the ATCC and screened for mycoplasma presence, showing no contamination. HEK293 cells were grown in Dulbecco's Modified Eagle Medium (DMEM, Thermo Fisher Scientific) supplemented with 10% (v/v) FBS, 50 μg/mL penicillin and 50 μg/mL streptomycin. Expression plasmids were transfected using Lipofectamine LTX Transfection Reagent (Thermo Fisher Scientific). USP9X plasmid (pCMV-HA-USP9X (DU10171)) was obtained through the MRC PPU Reagents and Services facility (MRC PPU, College of Life Sciences, University of Dundee, Scotland, mrcppureagents.dundee.ac.uk). Fat4 plasmids (pcDNA Fat4[ICD]::FLAG and pCMV5 Fat4[ΔECD]::FLAG) were a kind gift from Helen McNeill (Washington University School of Medicine, St. Louis). The pCMV-FLAG-EGFP plasmid was a kind gift from Nic Tapon (Francis Crick Institute, London).

## RNAi production and treatment

dsRNAs were synthesised using the Megascript T7 kit (Thermo Fisher Scientific) according to the manufacturer's instructions. DNA templates for dsRNA synthesis were PCR amplified from genomic DNA or plasmids encoding the respective genes using primers containing the 5′ T7 RNA polymerase-binding site sequence. dsRNA primers were designed using the DKFZ E-RNAi design tool (https://www.dkfz.de/signaling/e620rnai3/). The following primers were used: *lacZ* (Fwd –TTGCCGGGAAGCTAGAGTAA and Rev – GCCTTCCTGTTTTTGCTCAC); *faf* (Fwd – CATGCGCGTTTAGGCGAGTA and Rev – CGCACCACGCTGATGAGTA). After cell seeding, S2 cells were incubated with 20 μg dsRNA for 1 h in serum-free medium, before complete medium was added. 72 h after dsRNA treatment, cells were lysed and processed as detailed below.

## RNA isolation and RT-PCR analysis

For *faf* expression analysis in *UAS-faf* transgenes, *hsFLP; UAS-lacZ^{RNAi}; Act>CD2>RFP* females were crossed with control flies (*UAS-mCherry*), *UAS-faf^{isoC}* or *UAS-faf^{WT}*. 48 h after egg laying (AEL) developing larvae were heat-shocked at 37°C for 1 h. Late L3 larvae were collected, homogenised using disposable pellet pestles and total RNA was extracted using the QIAshredder and RNeasy kits (QIAGEN), according to the manufacturer's instructions. 1 μg of total RNA was used for cDNA production using the QuantiTect Reverse Transcription kit (QIAGEN), following the manufacturer's instructions. cDNA was used to conduct RT-PCR analysis using the following primers: *rp49* (Fwd – GACGCTTCAAGGGACAGTATCTG and Rev – GCAGTAAACGCGGTTCTGCATGAG); *faf* (Fwd – TAATACGACTCACTATAGGGACGCCAGAGCAAATGTTTTT and Rev – TAATACGACTCACTATAGGGACATGCTAAAGTCTTGCCCG). RT-PCR reactions were analysed in a 2% agarose gel and imaged in a Chemidoc MP Imaging System (Bio-Rad).

## Immunoprecipitation and immunoblot analysis

For purification of FLAG-tagged proteins, cells were lysed in lysis buffer (50 mM Tris pH 7.5, 150 mM NaCl, 1% Triton X-100, 10% (v/v) glycerol, and 1 mM EDTA), to which 0.1 M NaF, phosphatase inhibitors 2 and 3 (Sigma) and protease inhibitor cocktail (Complete, Roche) were added. Cell extracts were spun at 17,000 g for 10 min at 4 °C. FLAG-tagged proteins were purified using anti-FLAG M2 Affinity agarose gel (Sigma) for >1 h at 4 °C. FLAG immunoprecipitates were then washed four times with lysis buffer before elution using 150 ng/μl 3x FLAG peptide for 15–30 minutes at 4 °C. Detection of purified proteins and associated complexes was performed by immunoblot analysis using chemiluminescence (Thermo Fisher Scientific). Western blots were probed with mouse anti-FLAG (M2; Sigma; RRID:AB_262044), rat anti-HA (3F10; Roche Applied Science; RRID:AB_2314622), mouse anti-V5 (Thermo Fisher Scientific; RRID:AB_2556564), or mouse anti-tubulin (E7; DSHB; RRID:AB_528499). Secondary antibodies used included HRP-conjugated sheep anti-mouse (Amersham) and HRP-conjugated goat anti-rat (Thermo Fisher Scientific). For densitometry analysis of immunoblots, X-ray blots were scanned using an Epson Perfection V700 flatbed scanner and further analysed with the Gel Analyser function on ImageJ (RRID:SCR_003070). Alternatively, immunoblots were analysed in an ImageQuant 600 (Amersham).

## Immunostaining

Larval tissues were processed as previously described[71]. Primary antibodies were incubated overnight at 4°C unless otherwise stated. Mouse anti-Armadillo (N2 7A1; DSHB; RRID:AB_528089) was used at 1:50, rat anti-Fat (kind gift from Helen McNeill) was used at 1:500 and mouse anti-β-galactosidase (Z3781, Promega; RRID:AB_430877) was used at 1:500. Anti-mouse Rhodamine Red-X-conjugated (Jackson ImmunoResearch) secondary antibodies were used at 1:500. Anti-mouse or anti-rat Alexa Fluor 647-conjugated (Jackson ImmunoResearch) secondary antibodies were used at 1:500. Secondary antibodies were incubated for at least 2 h at room temperature. After washes, tissues were stained with DAPI (1μg/mL) for 10 minutes before clearing in Vectashield (without DAPI) (H-1200, Vector Labs; RRID:AB_2336790, respectively), and mounting with Mowiol 40-88 (Sigma). Fluorescence images were acquired on Zeiss LSM710 or Zeiss LSM880 confocal laser scanning microscopes (40x or 63x objective lens).

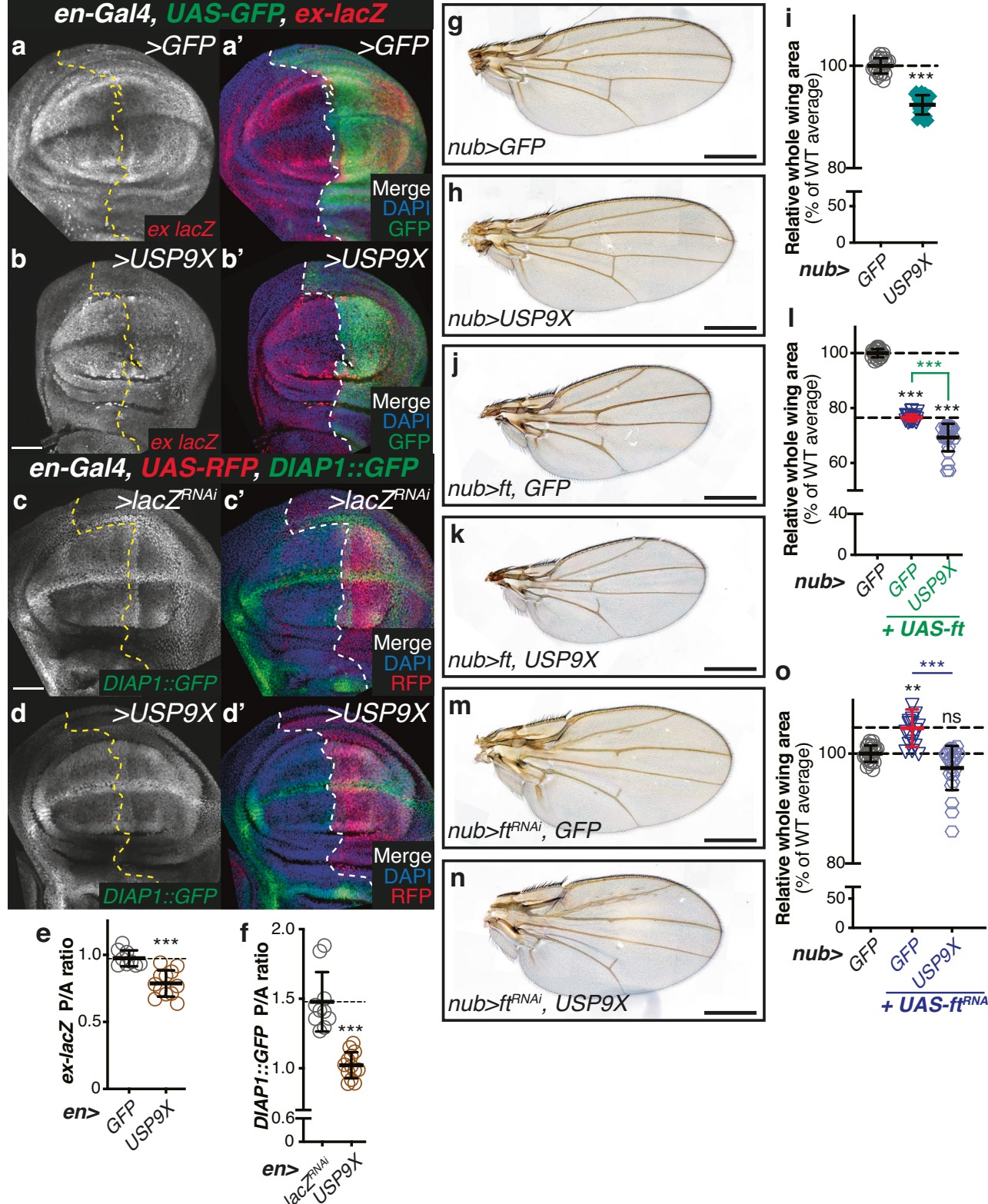

## Mammalian cell immunofluorescence

HEK293 cells were seeded onto glass coverslips coated with Poly-L-Lysine (Sigma) and left to adhere overnight prior to transfection with plasmid DNA. 24-48 h after transfection, coverslips were washed in PBS, fixed with 4% Paraformaldehyde and permeabilised in 0.1% Triton-X-100 in PBS before incubating with the primary antibody Rabbit anti-Fat4 (PA5-116735; Thermo Fisher Scientific; RRID: AB_2901366) at a

concentration of 1:100. Donkey anti-rabbit Rhodamine Red X-conjugated secondary antibody (Jackson Immunoresearch) was used at a concentration of 1:500. After washes, the cells were stained with DAPI (1μg/mL) for 5 minutes. Coverslips were then rinsed in distilled $H_2O$, dried on tissue and mounted onto microslides containing a droplet of Mowiol 40-88 (Sigma). Slides were left to dry overnight before imaging on a LSM 880 confocal microscope (40x objective).

**Fig. 8 | Faf-mediated regulation of Hpo and Ft signalling is conserved.**
**a–d** USP9X regulates Hpo signalling readouts in vivo. XY confocal micrographs of third instar wing imaginal discs carrying *ex-lacZ* (**a**, **b**) or *DIAP1::GFP* (**c**, **d**) and expressing *GFP* (**a**), *lacZ^RNAi* (**c**) or *UAS-USP9X* under the control of *en-Gal4* (b,d), showing β-Gal antibody staining (**a**, **b**; red in a', b') or direct GFP fluorescence (**c**, **d**; green in c', d'). Ventral is up in XY sections. GFP and RFP mark the *hh-Gal4-*expressing compartment (right) in a, b and c, d, respectively. Dashed lines depict anterior-posterior compartment boundary. Scale bar: 50 μm. **e**, **f** Quantification of Yki-mediated transcriptional reporter expression. Shown are posterior/anterior (P/A) ratios for *ex-lacZ* (**e**) and *DIAP1::GFP* (**f**) levels. Data are shown as average ± standard deviation, with all data points represented. (n = 9, 12 in e, and n = 10, 13 in **f**). Significance was assessed using an unpaired two-tailed t-test, with Welch's

correction. ***, p < 0.001. **g–o** USP9X regulates tissue growth and modulates Ft-mediated growth phenotypes. Shown are adult wings from flies raised at 25°C expressing the indicated transgenes in the wing pouch under the control of *nub-Gal4* (*nub >* ). Quantification of the effects of USP9X on growth is shown (**i**, **l** and **o**). Data are represented as % of the average wing area of controls (*nub > GFP*, average set to 100%). Data are shown as average ± standard deviation, with all data points depicted. (n = 22, 24 in l; n = 22, 21, 24 in l; and n = 22, 19, 23 in o). Significance was assessed using one-way ANOVA analysis comparing all genotypes to the *nub > GFP* control (black asterisks), with Dunnett's multiple comparisons test. For pairwise comparisons, unpaired two-tailed t-tests with Welch's correction were used. **,  p < 0.01; ***, p < 0.001; ns, non-significant. Scale bar: 500 μm.

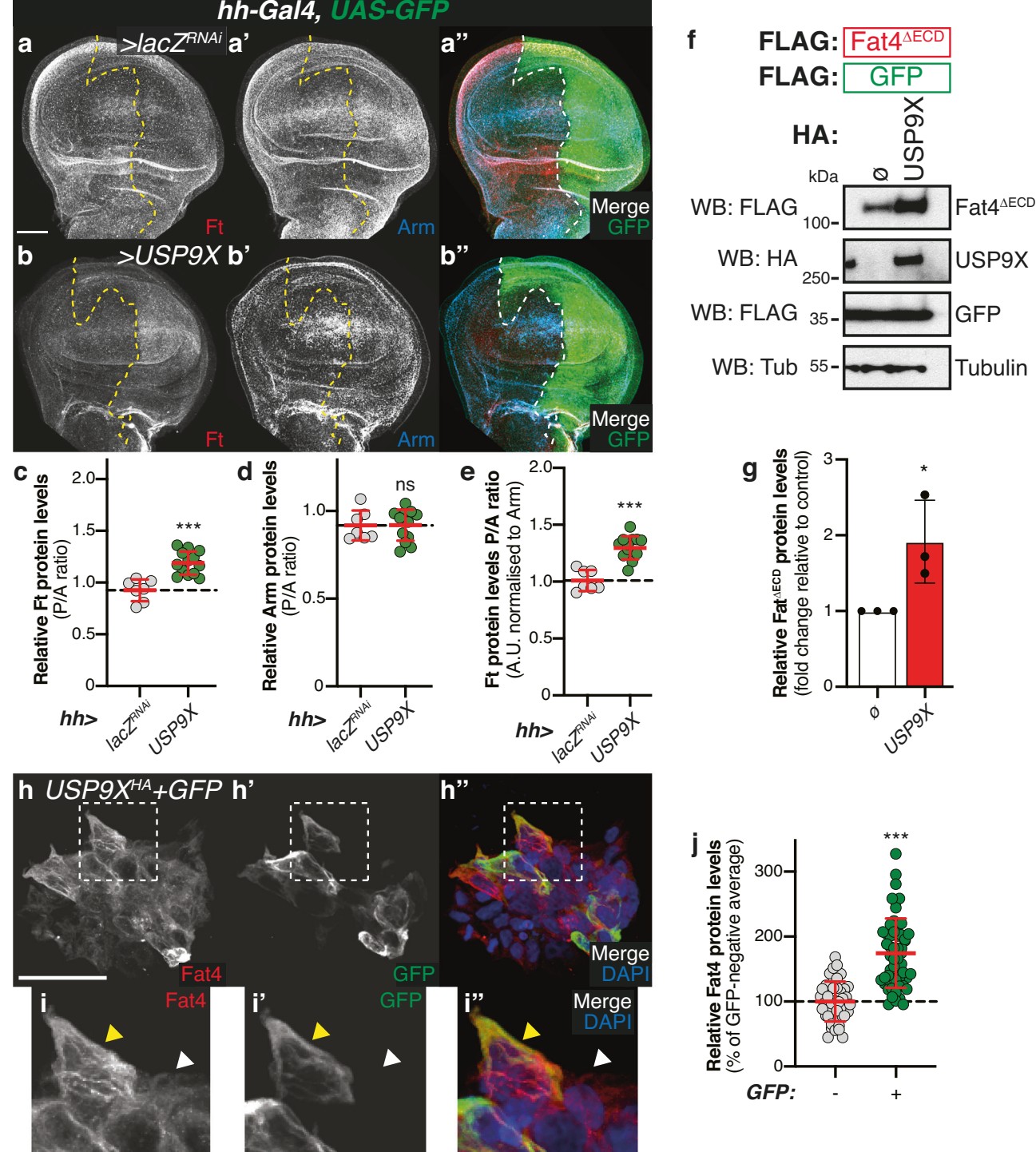

**Fig. 9 | USP9X regulates Ft protein levels. a, b** USP9X regulates Ft protein levels in vivo. XY confocal micrographs of third instar wing imaginal discs expressing the indicated constructs using *hh-Gal4*, showing Ft (**a**, **b**; red in **a"**, **b"**), or Arm antibody staining (**a'**, **b'**; cyan in **a"**, **b"**), and direct GFP fluorescence (green in **a"**, **b"**). Note that panels a-a" are identical to Figs. 7a–7a". Ventral is up in XY sections, whilst GFP marks the *hh-Gal4*-expressing compartment (right). Dashed white line depicts anterior-posterior compartment boundary. Scale bar: 50 μm. **c–e** Quantification of in vivo Arm and Ft protein levels. Shown are posterior/anterior (P/A) ratios for Fat (**c**), or Arm protein levels (**d**), as well as the normalised Ft protein levels (e, normalised to Arm). Data are shown as average ± standard deviation, with all data points represented. (*n* = 7, 13). Significance was assessed using an unpaired two-tailed t-test, with Welch's correction. ns, non-significant; ***, *p* < 0.001. **f**, **g** USP9X regulates Fat4 protein levels. FLAG-tagged Fat4$^{\Delta ECD}$ was co-expressed with FLAG-tagged GFP and either empty vector (ø) or HA-tagged USP9X in HEK293 cells before immunoblot analysis using the indicated antibodies. FLAG-GFP and Tubulin were used as transfection and loading controls, respectively. **g** Quantification of effect of

USP9X on Fat4$^{\Delta ECD}$ protein levels. Shown are relative Fat4 protein levels (fold change relative to empty plasmid (ø; set to 1); Fat4 protein levels were normalised to Tubulin) quantified from Western blot experiments. Data are represented as average ± standard deviation, with all data points represented. (*n* = 3 independent experiments). Significance was assessed by an unpaired two-tailed t-test. *, *p* < 0.05. **h–j** Analysis of USP9X-dependent regulation of Fat4 protein levels via immunostaining. Shown are XY confocal images (**h**) and insets (**i**) of HEK293 cells transfected with HA-tagged USP9X and GFP depicting Fat4 antibody staining (**h** and **i**; red in **h"** and **i"**) or direct fluorescence from GFP (**h'** and **i'**; green in **h"** and **i"**). DAPI (blue in **h"** and **i"**) stains nuclei. White dashed boxes represent inset images **i**, **i'** and **i"**. Yellow and white arrowheads in inset images indicate GFP-positive and GFP-negative cells, respectively. Scale bar: 50 μm (**h**) and 10 μm (**i**). **j** Quantification of Fat4 protein levels. Data are represented as average ± standard deviation, with all data points represented (*n* = 42, 54). Significance was assessed by an unpaired two-tailed t-test with Welch's correction. ***, *p* < 0.001.

## *Drosophila* genetics and genotypes

Transgenic RNAi stocks were obtained from the Vienna *Drosophila* Resource Centre (VDRC; RRID:SCR_013805) and the Kyoto Stock Centre (DGRC; RRID:SCR_008469). Details of RNAi stocks used in the in vivo RNAi screens are detailed in Table S1. *ft$^{G-rv}$* and *ft$^8$* were obtained from Yanlan Mao (UCL). *en-Gal4, UAS-GFP, ex-lacZ (ex$^{697}$)* and *mirr-Gal4* were obtained fron Nic Tapon (Francis Crick Institute). *en-Gal4, HRE-diap1::GFP* was obtained from Jin Jiang (UT Southwestern). *Fbxl7* stocks were obtained from Barry Thompson (ANU, Canberra). *en-Gal4, UAS-RFP* was obtained from Jean-Paul Vincent (Francis Crick Institute). The *faf* fly stocks *faf$^{B3}$* (BL25100), *faf$^{BX3}$* (BL25101), *faf$^{BX4}$* (BL25107), *faf$^{FO8}$* (BL25108), and *UAS-faf* (BL25102) were obtained from Bloomington. *UAS-faf$^{WT}$*, *UAS-faf$^{C1677S}$* (*UAS-faf$^{CD}$*) and *UAS-USP9X* were a kind gift from Bassem A. Hassan (Paris Brain Institute, France) and have been previously described[36].

All crosses were raised at 25°C unless otherwise stated. Genotypes were as follows:

Figures 1a, 3a, 6f, 8g, S1d, S2a, S4a, S5e: *nub-Gal4, UAS-GFP / UAS-GFP*

Figure 1b, S1e, S2g: *nub-Gal4, UAS-GFP / +; + / UAS-faf$^{RNAi SSGD}$* (VDRC 2955GD)

Figures 1c, 6g, S1h: *UAS-faf$^{isoC}$ / +; nub-Gal4, UAS-GFP*

Figures 1d, 6k, 8j: *nub-Gal4, UAS-ft$^{HA}$ / UAS-GFP*

Figure 1e: *nub-Gal4, UAS-ft$^{HA}$ / +; + / UAS-faf$^{RNAi SSGD}$* (VDRC 2955GD)

Figures 1f, 6l: *UAS-faf / +; nub-Gal4, UAS-ft$^{HA}$ / +*

Figures 1g, 6p, 8m: *nub-Gal4, UAS-ft$^{RNAi}$* (VDRC 108863KK) */ UAS-GFP*

Figure 1h: *nub-Gal4, UAS-ft$^{RNAi}$* (VDRC 108863KK) */ +; + / UAS-faf$^{RNAi SSGD}$* (VDRC 2955GD)

Figures 1i, 6q: *UAS-faf / +; nub-Gal4, UAS-ft$^{RNAi}$* (VDRC 108863KK) */ +*

Figure 2b, S3a: *w$^{iso}$*

Figure 2c, S3b: *ft$^{G-rv}$ / ft$^8$*

Figure 2d: *ft$^{G-rv}$ / ft$^8$; + / faf$^{FO8}$* (BL25108)

Figure 2e: *ft$^{G-rv}$ / ft$^8$; + / faf$^{BX4}$* (BL25107)

Figure 2f, S3c: *ft$^{G-rv}$ / ft$^8$; + / faf$^{B3}$* (BL25100)

Figure 2g: *ft$^{G-rv}$ / ft$^8$; + / faf$^{BX3}$* (BL25101)

Figure 3b: *nub-Gal4, UAS-ds / UAS-GFP*

Figure 3c: *nub-Gal4, UAS-ds / UAS-faf$^{RNAi 79GD}$* (VRDC 30679GD)

Figure 3d: *nub-Gal4, UAS-ds / UAS-faf$^{RNAi KK}$* (VRDC 107716KK)

Figure 3e: *nub-Gal4, UAS-ds / +; + / UAS-faf$^{RNAi SSGD}$* (VDRC 2955GD)

Figure 3f: *UAS-faf$^{isoC}$ / +; nub-Gal4, UAS-ds / +*

Figure 3g: *nub-Gal4, UAS-Dlish$^{RNAi}$* (VDRC 104282KK) */ UAS-GFP*

Figure 3h: *nub-Gal4, UAS-Dlish$^{RNAi}$* (VDRC 104282KK) */ +; + / UAS-faf$^{RNAi SSGD}$* (VDRC 2955GD)

Figure 3i: *UAS-faf$^{isoC}$ / +; nub-Gal4, UAS-Dlish$^{RNAi}$* (VDRC 104282KK) */ +*

Figure 3j: *nub-Gal4, UAS-Dlish$^{RNAi}$* (VDRC 104282KK) */ UAS-ft$^{HA}$*

Figure 3k: *nub-Gal4, UAS-Dlish$^{RNAi}$* (VDRC 104282KK) */ UAS-ft$^{HA}$; + / UAS-faf$^{RNAi SSGD}$* (VDRC 2955GD)

Figure 3l: *nub-Gal4 / UAS-GFP; UAS-Fbxl7$^{GFP}$ / +*

Figure 3m: *nub-Gal4 / +; UAS-Fbxl7$^{GFP}$ / UAS-faf$^{RNAi SSGD}$* (VDRC 2955GD)

Figure 3n: *nub-Gal4 / UAS-ft$^{HA}$; UAS-Fbxl7$^{GFP}$ / +*

Figure 3o: *nub-Gal4 / UAS-ft$^{HA}$; UAS-Fbxl7$^{GFP}$ / UAS-faf$^{RNAi SSGD}$* (VDRC 2955GD)

Figures 4e, 7a, 9a: *+ / UAS-lacZ$^{RNAi}$; hh-Gal4, UAS-GFP / +*

Figure 4f: *hh-Gal4, UAS-GFP / UAS-faf$^{RNAi SSGD}$* (VDRC 2955GD)

Figure 4g: *UAS-faf$^{isoC}$ / +;; hh-Gal4, UAS-GFP / +*

Figure 4h: *UAS-ft$^{RNAi}$* (VDRC 108863KK) */ +; hh-Gal4, UAS-GFP / +*

Figures 5a, 6a, 8a: *en-Gal4, UAS-GFP, ex-lacZ (ex$^{697}$) / UAS-GFP; MKRS / +*

Figure 5b: *en-Gal4, UAS-GFP, ex-lacZ (ex$^{697}$) / UAS-hpo$^{RNAi}$* (VDRC 104169KK); *MKRS / +*

Figure 5c: *en-Gal4, UAS-GFP, ex-lacZ (ex$^{697}$) / +; MKRS / UAS-faf$^{RNAi SSGD}$* (VDRC 2955GD)

Figure 5d: *en-Gal4, UAS-GFP, ex-lacZ (ex$^{697}$) / UAS-ft$^{HA}$; MKRS / +*

Figure 5e: *en-Gal4, UAS-GFP, ex-lacZ (ex$^{697}$) / UAS-ft$^{HA}$; MKRS / UAS-faf$^{RNAi SSGD}$* (VDRC 2955GD)

Figures 5g, 8c, S9a: *en-Gal4, UAS-RFP / UAS-lacZ$^{RNAi}$; HRE-diap1::GFP / +*

Figure 5h: *en-Gal4, UAS-RFP / UAS-hpo$^{RNAi}$* (VDRC 104169KK); *HRE-diap1::GFP / +*

Figure 5i: *en-Gal4, UAS-RFP / +; HRE-diap1::GFP / UAS-faf$^{RNAi SSGD}$* (VDRC 2955GD)

Figure 5j: *en-Gal4, UAS-RFP / UAS-ft$^{HA}$; HRE-diap1::GFP / +*

Figure 5k: *en-Gal4, UAS-RFP / UAS-ft$^{HA}$; HRE-diap1::GFP / UAS-faf$^{RNAi SSGD}$* (VDRC 2955GD)

Figure 6b: *UAS-faf$^{isoC}$ / +; en-Gal4, UAS-GFP, ex-lacZ (ex$^{697}$) / UAS-GFP; MKRS / +*

Figure 6c: *en-Gal4, UAS-GFP, ex-lacZ (ex$^{697}$) / UAS-GFP; MKRS / UAS-faf$^{WT}$*

Figure 6d: *en-Gal4, UAS-GFP, ex-lacZ (ex$^{697}$) / UAS-GFP; MKRS / UAS-faf$^{CD}$ (faf$^{C1677S}$)*

Figure 6h: *nub-Gal4, UAS-GFP / +; + / UAS-faf$^{WT}$*

Figure 6i: *nub-Gal4, UAS-GFP / +; + / UAS-faf$^{CD}$ (faf$^{C1677S}$)*

Figure 6m: *nub-Gal4, UAS-ft$^{HA}$ / +; + / UAS-faf$^{WT}$*

Figure 6n: *nub-Gal4, UAS-ft$^{HA}$ / +; + / UAS-faf$^{CD}$ (faf$^{C1677S}$)*

Figure 6r: *nub-Gal4, UAS-ft$^{RNAi}$* (VDRC 108863KK) */ +; + / UAS-faf$^{WT}$*

Figure 6s: *nub-Gal4, UAS-ft$^{RNAi}$* (VDRC 108863KK) */ +; + / UAS-faf$^{CD}$ (faf$^{C1677S}$)*

Figure 7b: *hh-Gal4, UAS-GFP / UAS-faf$^{WT}$*

Figure 7c: *hh-Gal4, UAS-GFP / UAS-faf$^{CD}$ (faf$^{C1677S}$)*

Figure 8b: *en-Gal4, UAS-GFP, ex-lacZ (ex$^{697}$) / UAS-GFP; MKRS / UAS-USP9X*

Figure 8d: *en-Gal4, UAS-RFP / +; HRE-diap1::GFP / UAS-USP9X*

Figure 8h: *nub-Gal4, UAS-GFP / +; + / UAS-USP9X*
Figure 8k: *nub-Gal4, UAS-ft^HA / +; + / UAS-USP9X*
Figure 8n: *nub-Gal4, UAS-ft^RNAi (VDRC 108863KK) / +; + / UAS-USP9X*
Figure 9b: *hh-Gal4, UAS-GFP / UAS-USP9X*
Figure S1f: *nub-Gal4, UAS-GFP / UAS-faf^RNAi KK (VDRC 107716KK)*
Figure S1g: *nub-Gal4, UAS-GFP / UAS-faf^RNAi 79GD (VDRC 30769GD)*
Figure S4b: *nub-Gal4, UAS-ex^2 / UAS-GFP*
Figure S4c: *nub-Gal4, UAS-ex^2 / +; + / UAS-faf^RNAi SSGD (VDRC 2955GD)*
Figure S4d: *nub-Gal4, UAS-ex^2 / UAS-ft^HA*
Figure S4e: *nub-Gal4, UAS-ex^2 / UAS-ft^HA; + / UAS-faf^RNAi SSGD (VDRC 2955GD)*
Figure S4f: *nub-Gal4, UAS-elgi^RNAi (VDRC 109617KK) / UAS-GFP*
Figure S4g: *nub-Gal4, UAS-elgi^RNAi (VDRC 109617KK) / +; + / UAS-faf^RNAi SSGD (VDRC 2955GD)*
Figure S4h: *UAS-faf^isoC / +; nub-Gal4, UAS-elgi^RNAi (VDRC 109617KK) / +*
Figure S4i: *nub-Gal4, UAS-elgi^RNAi (VDRC 109617KK) / UAS-ft^HA*
Figure S4j: *nub-Gal4, UAS-elgi^RNAi (VDRC 109617KK) / UAS-ft^HA; + / UAS-faf^RNAi SSGD (VDRC 2955GD)*
Figure S4k: *nub-Gal4, UAS-app^RNAi (VDRC 32863GD) / UAS-GFP*
Figure S4l: *nub-Gal4, UAS-app^RNAi (VDRC 32863GD) / +; + / UAS-faf^RNAi SSGD (VDRC 2955GD)*
Figure S4m: *UAS-faf^isoC / +; nub-Gal4, UAS-app^RNAi (VDRC 32863GD) / +*
Figure S4n: *nub-Gal4, UAS-app^RNAi (VDRC 32863GD) / UAS-ft^HA*
Figure S4o: *nub-Gal4, UAS-app^RNAi (VDRC 32863GD) / UAS-ft^HA; + / UAS-faf^RNAi SSGD (VDRC 2955GD)*
Figure S5a: *nub-Gal4, UAS-hpo^RNAi (VDRC 104169KK) / UAS-GFP* (Room temperature cross)
Figure S5b: *UAS-faf^isoC / +; nub-Gal4, UAS-hpo^RNAi (VDRC 104169KK) / +* (Room temperature cross)
Figure S5c: *nub-Gal4, UAS-hpo^RNAi (VDRC 104169KK) / +; + / UAS-faf^RNAi SSGD (VDRC 2955GD)* (Room temperature cross)
Figure S5f: *nub-Gal4 / UAS-GFP; + / UAS-wts*
Figure S5g: *UAS-faf^isoC / +; nub-Gal4 / +; UAS-wts / +*
Figure S5h: *nub-Gal4 / +; UAS-wts / UAS-faf^RNAi SSGD (VDRC 2955GD)*
Figure S5i: *nub-Gal4 / UAS-ft^HA; UAS-wts / +*
Figure S5j: *nub-Gal4 / UAS-ft^HA; UAS-wts / UAS-faf^RNAi SSGD (VDRC 2955GD)*
Figure S7a: *UAS-ft^HA / +; hh-Gal4, UAS-GFP / +*
Figure S7b: *UAS-ft^HA / +; hh-Gal4, UAS-GFP / UAS-faf^RNAi SSGD (VDRC 2955GD)*
Figure S7g: *UAS-GFP / UAS-GFP; mirr-Gal4 / +*
Figure S7h: *UAS-GFP / +; mirr-Gal4 / UAS-faf^WT*
Figure S7i: *UAS-GFP / +; mirr-Gal4 / UAS-faf^RNAi SSGD (VDRC 2955GD)*
Figure S8b, S8h, S9f, S9i, S10d, S10f: *en-Gal4, UAS-RFP / UAS-lacZ^RNAi; Dachs::GFP / +*
Figure S8c, S8i: *en-Gal4, UAS-RFP / UAS-ft^HA; Dachs::GFP / +*
Figure S8d, S8j: *en-Gal4, UAS-RFP / UAS-ft^HA; Dachs::GFP / UAS-faf^RNAi SSGD (VDRC 2955GD)*
Figure S8e, S8k: *UAS-faf^isoC / +; en-Gal4, UAS-RFP / +; Dachs::GFP / +*
Figure S8f, S8l: *en-Gal4, UAS-RFP / UAS-ft^RNAi (VDRC 108863KK); Dachs::GFP / +*
Figure S8g, S8m: *en-Gal4, UAS-RFP / +; Dachs::GFP / UAS-faf^RNAi SSGD (VDRC 2955GD)*
Figure S9b: *UAS-faf^isoC / +; en-Gal4, UAS-RFP / +; HRE-diap1::GFP / +*
Figure S9c: *en-Gal4, UAS-RFP / +; HRE-diap1::GFP / UAS-faf^WT*
Figure S9d: *en-Gal4, UAS-RFP / +; HRE-diap1::GFP / UAS-faf^CD (faf^C1677S)*
Figure S9g, S9j: *en-Gal4, UAS-RFP / UAS-lacZ^RNAi; Dachs::GFP / UAS-faf^WT*
Figure S9h, S9k: *en-Gal4, UAS-RFP / UAS-lacZ^RNAi; Dachs::GFP / UAS-faf^CD (faf^C1677S)*

Figure S9l: *hsFLP / +; UAS-lacZ^RNAi / UAS-mCherry; act > CD2>Gal4, UAS RFP / +* (mCherry sample)
Figure S9l: *hsFLP / UAS-faf^isoC; UAS-lacZ^RNAi / +; act > CD2>Gal4, UAS RFP / +* (faf^isoC sample)
Figure S9l: *hsFLP / +; UAS-lacZ^RNAi / +; act > CD2>Gal4, UAS RFP / UAS-faf^WT* (faf^WT sample)
Figure S10e, S10g: *en-Gal4, UAS-RFP / UAS-lacZ^RNAi; Dachs::GFP / UAS-USP9X*

**Immunofluorescence quantification and statistical analyses**
For quantification of Ft and Arm protein levels in vivo (Figs. 7d-f, 9c-e, S7c-e), relative *ex-lacZ* (Figs. 5f, 6e and 8e) or *diap1::GFP* levels (Figs. 5l, 8f and S9e), ratios in posterior versus anterior compartment (P/A) were calculated by manually drawing around each compartment of the wing disc in maximum intensity projections, then measuring the mean grey pixel value in Fiji (RRID:SCR_002285). For quantification of Ft protein levels in eye imaginal discs (Figure S7f), ratios between GFP-positive and GFP-negative areas were calculated by manually drawing around the relevant sections of the eye disc in maximum intensity projections and measuring mean grey pixel values in Fiji. Significance was calculated using a Brown-Forsythe and Welch ANOVA comparing all genotypes to their respective controls (for Ft and Arm protein levels), or a one-way ANOVA comparing all means to the *en-Gal4* control (*UAS-GFP* for *ex-lacZ*; *UAS-lacZ^RNAi* for *diap1::GFP*), both with Dunnett's multiple comparisons test. Pairwise comparisons between other samples were performed using unpaired two-tailed t-tests or unpaired two-tailed t-tests with Welch's correction.

**Analysis of larval wing imaginal disc sizes**
For tissue size analysis of *ft* trans-heterozygous mutants, crosses were setup at 25°C and flies were placed in new vials every 24 h for precise staging of larval development. Larvae were dissected at the late L3 stage, ~5 days AEL.

**Analysis of genetic interactions in *Drosophila* adult wings**
For analysis of genetic interactions in the *Drosophila* wing, flies with the genotypes of interest were collected and preserved in 70% EtOH for at least 24 h. Adult wing samples were prepared as previously described[72]. Briefly, wings were removed in 100% isopropanol, mounted in microscope slides using Euparal (Anglian Lepidopterist Supplies) as mounting medium and baked at 65°C for at least 5 h. Adult wing images were captured using a Pannoramic 250 Flash High Throughput Scanner (3DHISTECH) and extracted using the Pannoramic Viewer software (3DHISTECH) or captured using an Axioscan 7 (Zeiss). Wing area was quantified using Fiji (the alula and costal cell of the wing were both excluded from the analysis). Images were processed using Adobe Photoshop (RRID:SCR_014199).

**Analysis of planar cell polarity phenotypes**
For analysis of tissue roundness (Fig. 1k, S2c, S2d), images of adult wings were analysed in Fiji and the ratio between the proximal-distal (PD) axis (defined by a line that bisects the area between the L3 and L4 veins in half) and the anterior-posterior axis (defined by a line that crosses the L4/L5 crossvein and generally defines the widest part of the adult wing) was calculated (Figure S2b). Additionally, circularity was calculated using the Fiji area selection tool to define the adult wing, or by fitting an ellipse to the wing blade section (Figure S2b'). For analysis of wing hair orientation (Figure S2i), adult wing images were used to define 3 regions of interest (ROI) within the section of the wing between the L3 and L4 veins. ROIs were used to train a model to extract features in Ilastik (RRID:SCR_015246) by defining specific pixels as part of a wing hair or as part of the wing background. Images corresponding to the probability model from Ilastik[73] were analysed in Fiji, thresholded to include only the wing hair and wing hair orientation was

quantified using OrientationJ (RRID:SCR_014796) in Fiji[74]. Data was plotted as normalised pixel frequencies (0 to 3.5) for each 0.5° angle and 180° rose plots were generated using R studio.

## Analysis of wing vein defects

For analysis of L2 vein defects in adult wings (Figure S2h), images of adult wings were analysed for the presence or absence of vein defects and the phenotype frequency was calculated. Significance was calculated in Prism using Chi-square analysis, adjusted Fisher's exact test and Bonferroni correction for multiple comparisons.

## Statistical analysis

Statistical analyses were performed using GraphPad Prism 8, 9 and 10 (RRID:SCR_002798). Significance values corresponding to comparisons between two groups were calculated using unpaired t-test or unpaired t-test with Welch's correction. Significance values corresponding to comparisons between multiple groups were calculated using one-way ANOVA, Brown-Forsythe and Welch ANOVA or Kruskal-Wallis ANOVA analyses with Dunnett's, Tukey's or Dunn's multiple comparisons tests. $p$-values: ns denotes not significant; * denotes $I < 0.05$; ** denotes $p < 0.01$; *** denotes $p < 0.001$. Significance values are provided in Table S2.

## Reporting summary

Further information on research design is available in the Nature Portfolio Reporting Summary linked to this article.

## Data availability

Source data are provided with this paper.

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

## Acknowledgements

We thank the Vienna *Drosophila* Resource Centre and the Bloomington *Drosophila* Stock Centre (NIH P40OD018537) for transgenic fly stocks. The antibodies E7 (DSHB Hybridoma Product E7), 2A1 (DSHB Hybridoma Product 2A1) and N2 7A1 (DSHB Hybridoma Product N2 7A1 Armadillo) were deposited by M. Klymkowsky, R. Holmgren and E. Wieschaus to the Developmental Studies Hybridoma Bank, created by the NICHD of the NIH and maintained at The University of Iowa, Department of Biology. We thank N. Tapon, H. McNeill, B. Thompson, Y. Mao, J. P. Vincent and B. Hassan for providing fly stocks; Sam Wallis and Linda Hammond for assistance with microscopy; and C. Mohanathas for assistance in preliminary experiments. We thank members of the Ribeiro lab for helpful discussions and N. Tapon, M. Holder, S.A. Martin, and H. McNeill for critical reading of the manuscript. This work was supported by funding

from Cancer Research UK (C16420/A18066), The Academy of Medical Sciences/Wellcome Trust Springboard Award (SBF001/1018), Barts Charity (G-002400) and from the Biotechnology and Biological Sciences Research Council (BB/T004576/1). L.E.D. was supported by a Biotechnology and Biological Sciences Research Council LIDo PhD studentship (BB/M009513/1). R.I.L. was supported by a Medical Research Council MRC-DTP PhD studentship (MR/W007045/1).

## Author contributions

P.S.R. conceived the project and supervised the study; P.S.R., L.E.D., A.S., A.D.F. designed the experiments; P.S.R., L.E.D., A.S., A.D.F., R.I.L., H.S.B performed the experiments and analysed the data; P.S.R., L.E.D., A.S. wrote the manuscript; P.S.R., L.E.D., A.S. contributed to manuscript editing.

## Competing interests

The authors declare no competing interests.
