## [Peer Review file · Nature Communications]

The deubiquitylating enzyme Fat Facets promotes Fat signalling and restricts tissue growth

Corresponding Author: Dr Paulo Ribeiro

Version 0:

Reviewer comments:

Reviewer #1

(Remarks to the Author)

This manuscript describes the novel interaction of the Drosophila cell adhesion molecule Fat and the DUB, Fat facets (Faf), and a role for Faf in Fat pathway growth control. Previous studies have established Fat as a major regulator of tissue growth and patterning. Here, the authors conduct an RNAi screen against Drosophila DUB, and identify Faf as a regulator of wing growth. They show that fat facets binds to Fat, and to the Fat- ICD, and altering Fat facets levels alters Fat protein levels, suggesting that Faf acts to deubiquitinate Fat and thereby stabilize it. They provide evidence that Faf catalytic function is needed for its activity in the Fat pathway by using catalytically inactive Faf in some experiments. There is abundant genetic interaction data to support the idea that Faf acts in a Fat/Hippo pathway, and the finding that Faf binds to Fat, and specifically the Fat ICD is interesting and novel. It is also interesting that the mammalian homologs interact, and that USP9X stabilizes Fat4, suggesting this is an important interaction that has been conserved. However many of the figures in this manuscript are difficult to see, and overall quantification of many of the images is necessary.

Specific comments and suggestions

More information should be provided in the introduction and discussion about the Faf homologue, Usp9x and the known roles of Faf—and the complexity of the many possible targets of Faf.

For the non-Drosophila reader- it would help to clarify Ex-lacZ and Diap1-GFP as Hippo read-outs. (Line 195).

Figure 2. shows that there is a synergy between loss of Fat and loss of Fat facets

Panel H would be easier to read if genotypes were on the X axis.

Overall the genetic data presented does not clarify if fat and faf act in a single pathway or in parallel. In this reviewers opinion the FatGRV/Fd is likely a null, so the enhancement suggests a parallel pathway

Figure 3 shows more genetic interactions with Ds, Dlish and Fbx17 generally support the concept that Faf interacts with the Fat-Hippo pathway.

Figure 4 is not very clear.

panel C is not clear -what are the bands on the left side of the gel?

Not clear how I and J is different – quantification would help to be convincing of a change in overexpressed Fat protein levels upon faf depletion.

Figure 5—D staining is difficult to see—quantification of levels and cytoplasmic/membrane ratios would help. Perhaps mosaics would make this clearer? Putting Armadillo in the middle makes it more difficult to compare directly the panels of D.

Figure 6. Overall the changes in-ex-lacZ in this figure are more clear than those of Diap-GFP. It is obvious that faf rna1 leads to increased ex lacz, and that overexpressed Ft decreases Hippo targets. Overexpression of Fat and knockdown of faf in the picture seems to be equal but in the quantification it is different. Perhaps the figure is not representative?

Fig 7 The use of Faf and faf wt is confusing. Does Faf-wt express more transcript? qPCR could help clarify that, if this is an important point.

Fig 8. Panel G shows *faf* stabilizes *Fat*. But is it 2x as shown in (H)? The quantification does not seem to reflect the gel shown.

Fig 9. Ex LacZ effect is clear but quantification of *diap1* is not clearly matching- perhaps the authors could comment on this?

It would strengthen the paper to show that *Fat* is ubiquitinated and to see if *Faf* affects the levels of *Fat* ubiquitination, in the S2 cell system. However, this is not essential.

Similarly, to be confident that PCP is not affected the authors could examine wing hair orientation or ridges. Quantification of roundness could be in main figures

Reviewer #2

(Remarks to the Author)

This manuscript identified the deubiquitination enzyme *Fat* facets (*Faf*) as a positive regulator of *Fat* (*Ft*) function in controlling organ growth. *Ft* has been well characterized as an upstream activator of Hippo signaling, however its regulation by *Faf* has not been reported. This study fills this gap and advances our understanding of how ubiquitination controls Hpo signaling via the *Faf/Ft* axis. The experiments are logically and clearly presented, and the data are generally of high quality. *Faf* and *Ft* interact genetically and physically, and both regulate organ growth and Hpo target gene expression. The effects of *Faf* on *Ft* function require catalytic activity. The *Faf/Ft* interaction appears to be conserved, as the key interactions are recapitulated by using mammalian orthologs.

The manuscript can be strengthened by addressing the following:

Major:

1. There were other DUBs identified in the initial screen (Fig. S1) that partially suppressed *Ft*-mediated wing undergrowth phenotype. Why was *Faf* selected for further study? Do the other DUBs also contribute to *Ft* regulation, i.e. does their perturbation also affect *Ft* function?
2. Does *faf* genetically interact with Hippo pathway components other than *Ft* and its immediate network (e.g. *Hpo*, *wts*, *yki*, *sd*)? In other words, are the effects going through Hpo core components? This is important to ascertain in order to better place *Faf* in the pathway.
3. The story would benefit from a more rigorous demonstration of the effects of *Faf* on *Ft* protein levels. Since demonstration of decreased *Ft* ubiquitination in the presence of *Faf* appears technically challenging (as stated in Discussion), other molecular assays could be used. For example, at least the effects of proteasome inhibitors should be tested, and also *Ft* stability with and without *Faf* can be tested in a cycloheximide chase assay.
4. The specificity of the DUB inhibitor WP1130 should be described in the text. How specific is it for *Faf*? Is it possible that it inhibits other DUBs?

Minor:

1. Was modulation of the L2 vein phenotype in Fig S2I significant? Statistical assessment should be provided.
2. Effects on D (Fig 5 and other figures) are not particularly striking. Maybe use enlargements to show effects in single cells, and point out the differences?
3. Fig 5E panels are not referred to in the text.
4. Refer to UAS-*faf* and UAS-*faf*WT differently – both transgenes express wild type *Faf* protein, so a differentiating nomenclature (e.g. based on insertion site, etc.) should be used to avoid confusion.
5. Line 451, “*Faf* directly affected Hippo signalling readouts in vivo” – unclear what this means. There was an effect on Hpo target genes, but the word “directly” should be dropped, because obviously *Faf* does not transcriptionally regulate Hpo reporter genes.

Reviewer #3

(Remarks to the Author)

The manuscript by Dawson et al., explores roles for ubiquitination in the Hippo signalling pathway in *Drosophila*, with a focus on the *Fat* cadherin branch of the pathway. They perform a small scale in vivo genetic screen and identify the deubiquitylating enzyme *Fat* facets (*Faf*) as a positive regulator of *Fat* signalling in the context of tissue growth control. This adds new information to Hippo/*Fat* signalling. In general, the study is high on technical rigor – i.e. the wing growth experiments, immunofluorescence and biochemistry. The data are mostly convincing, well-controlled and rigorous, although some alternative experimental approaches and further quantification are required to fully prove the reported role of *Faf*. This is important as some of the effects are quite subtle - e.g. the impact of *Faf* depletion on *Fat* abundance, *Dachs* localisation and *Yki* activity. The subtle nature of the impact of *Faf* on *Fat* is reflected in the fact that the authors build their case through mass action (ten figures and many experiments) as opposed to a smaller set of very convincing results. The experiments on human USP9X and *Fat4* do not add much value to the paper especially because they do not further clarify the role of *Fat* homologues in Hippo signalling in mammals, which is still very controversial.

The major thing that is currently lacking from the manuscript is a precise understanding of the relative importance of ubiquitination to *Fat*/Hippo signalling and growth. More specifically:

- when and where does Fat Facets, and ubiquitination more broadly, influence Fat/Hippo signalling in growing tissues?
- does it have a major impact or a minor impact on Fat/Hippo signalling?
- Does it happen at specific stages of development and/or on other timescales, like specific periods of the cell cycle?
- Or does ubiquitination have special roles when cells encounter specific cues?
- Alternatively, does ubiquitination help set baselines or thresholds of Fat/Hippo signalling that are then modulated by other things like mechanical forces, or something else?

If the above questions cannot be addressed experimentally, they should at least be discussed. Currently, the Discussion is very long and focussed on molecular aspects of Faf and Fat. A more rounded Discussion of Faf would enhance the manuscript; some space for this could be made by removing repetition at the start of the Discussion, which recaps the introduction and results sections.

Other things:

Figure 2

I may have missed it, but I could not find information on the disc size experiments in the results, legends or methods. Specifically, were the animals carefully staged and dissected at the same age? This is essential because fat loss substantially extends the larval period of development for several days and thus gives imaginal discs an extended growth period. Different results are possible depending on when the tissues are removed and analysed.

Figure 4

The Fat abundance result in Figure 4 is very important and looks to be very modestly different if at all in 4F with Faf RNAi. This important result should be validated with faf mutant allele clones and high resolution/high magnification images should be presented if the result is consistent.

Also, the current data here should be quantified across multiple tissues and high-resolution images should be presented in the main figure or the supp figures. This is done for the USP9X experiments in Fig 10 but not in Fig 4.

Finally, what is the prominent D/V band in 4F and 4H? This looks very different to the Fat stains in other figure panels and in published studies. It looks like Wingless protein and is distracting.

Figure 5

The Dachs data are very difficult to follow. First, the inclusion of the 4 different gray panels per row that look very similar does not make it easy for readers to know what they are looking at. Second, the reported changes in Dachs are not readily appreciable by eye and appear to be very subtle. Dachs polarisation is easier to appreciate when done in a clonal fashion – e.g., PMID: 18694569. Can this clonal approach be used here to test Faf's role in Dachs localisation more convincingly? Alternatively, the Arm data could be removed to supps and high resolution images of Dachs shown. Is there any quantification?

The impact of Faf on Fat/Dachs could be further validated by exploring Vamana localisation, which is also sensitive to Fat/Dachs levels.

Figure 6

Assessment of Yki activity is another important experiment and currently relies on RNAi. It should also be performed in faf mutant allele clones, especially because the RNAi results show signification elevation one Yki target gene but not he other. Also, looking at the data it appears that there are region-specific changes in Yki target genes. This could be teased out further with the mutant clone approach.

Version 1:

Reviewer comments:

Reviewer #1

(Remarks to the Author)

This manuscript nicely shows the novel interaction between Ft and Fat Facets, and further shows that Fat facets regulates Fat levels and growth signaling, via genetic interactions and supports biochemical interactions. This is a nice addition to our understanding of growth regulation and the Ft pathway, as well as adding to our understanding of DUBs in growth regulation.

Fat facets (Faf) is a DUB, and the authors provide evidence that Faf catalytic function is needed for its activity in the Fat pathway by using catalytically inactive Faf in some experiments. They also show that USP9X stabilizes Fat4, suggesting this is an important interaction that has been conserved.

In the revised manuscript, the authors have tried to provide further proof that D is polarized, however without mosaic analysis this is still unconvincing, and should be removed or put in the supplemental materials. However I do not think this is required for publication, and in fact including this in the manuscript weakens the story.

Similarly, Figure 4i and 4J are not needed, and are better put in supplemental.

Reviewer #2

(Remarks to the Author)

The authors have successfully addressed my concerns.

Reviewer #3

(Remarks to the Author)

The authors have worked hard to respond to the comments from the three reviewers and the manuscript has been improved somewhat. The majority of new experiments on protein abundance have been done with S2 cells as opposed to in vivo in epithelial tissues. In addition, the authors have not introduced any clonal experiments in epithelial tissues with loss of function *faf* alleles. This is a significant weakness of the paper. The authors state they could not recombine a *faf* allele onto an FRT chromosome. This is very surprising and not insurmountable given other approaches could have been taken; with the advent of CRISPR/Cas9, it would be trivial for the authors to generate new *faf* loss of function alleles on an FRT82B chromosome, circumventing the need for recombination. I don't think the authors should have to do this now, but the lack of clonal experiments with *faf* loss of function alleles will reduce the confidence that readers will have in the paper and the likelihood of stimulating further studies of *faf* and the Hippo pathway by others.

We thank the reviewers for their insightful comments and for their suggestions which have helped us to substantially improve our manuscript. We summarise our main new findings before providing a point by point reply to the reviewers' comments. For some of the points, we also include extensive data in the rebuttal letter which, if so required by the reviewers, we can include in the revised manuscript.

Based on the reviewers' suggestions, we include the following new data in the revised manuscript:

- 1) **Figure 1k** and **Figure S2i** show data related to analysis of PCP phenotypes;
- 2) **Figure 4** now contains updated images for panels **c**, **f** and **h** for improved clarity;
- 3) **Figure 5** now contains all genotypes analysed and plot profiles for D and Arm localisation to highlight differences associated with modulation of Faf function;
- 4) **Figure 7** and **Figure 9** include plot profiles for D and Arm localisation to highlight differences;
- 5) **Figure 8g** includes new data to clarify effect of Faf catalytic activity in the regulation of Ft protein levels;
- 6) **Figure S2h** includes statistical analysis of effect of Faf on L2 wing vein defect;
- 7) **Figure S3** includes data regarding Ft localisation in *ft* trans-heterozygous animals;
- 8) **Figure S5** shows data related to the epistatic position of Faf within the Hpo signalling pathway;
- 9) **Figure S6** includes new data showing that Faf-mediated regulation of Ft protein levels also occurs in the eye imaginal disc;
- 10) **Figure S8** includes data regarding the relative expression levels of Faf transgenes;
- 11) **Figure S9** includes biochemical data regarding proteasome inhibition and DUB inhibition in the absence or presence of Faf.

Below, we provide further details and point by point responses to the reviewer's comments.

Reviewer #1 (Remarks to the Author):

This manuscript describes the novel interaction of the *Drosophila* cell adhesion molecule Fat and the DUB, Fat facets (Faf), and a role for Faf in Fat pathway growth control. Previous studies have established Fat as a major regulator of tissue growth and patterning. Here, the authors conduct an RNAi screen against *Drosophila* DUB, and identify Faf as a regulator of wing growth. They show that fat facets binds to Fat, and to the Fat-ICD, and altering Fat facets levels alters Fat protein levels, suggesting that Faf acts to deubiquitinate Fat and thereby stabilize it. They provide evidence that Faf catalytic function is needed for its activity in the Fat pathway by using catalytically inactive Faf in some experiments. There is abundant genetic interaction data to support the idea that Faf acts in a Fat/Hippo pathway, and the finding that Faf binds to Fat, and specifically the Fat ICD is interesting and novel. It is also interesting that the mammalian homologs interact, and that USP9X stabilizes Fat4, suggesting this is an important interaction that has been conserved. However many of the figures in this manuscript are difficult to see, and overall quantification of many of the images is necessary.

Response: We thank the reviewer for their comments and for the suggestions to improve the strength of our conclusions. We believe that the revised manuscript and the new data address the comments of the reviewer and further corroborate our findings.

1) More information should be provided in the introduction and discussion about the Faf homologue, Usp9x and the known roles of Faf – and the complexity of the many possible targets of Faf.

Response: We thank the reviewer for this suggestion, and we now include more information in the introduction regarding previously identified functions of Faf in *Drosophila*. The updated discussion now also includes extensive discussion of the potential roles of USP9X on the regulation of Hippo signalling.

2) For the non-*Drosophila* reader – it would help to clarify *Ex-lacZ* and *Diap1-GFP* as Hippo read-outs (Line 195).

Response: We have made the suggested changes to clarify that *ex-lacZ* and *diap1::GFP* are established Hippo pathway readouts (Line 268).

3) Figure 2 shows that there is a synergy between loss of Fat and loss of Fat facets. Panel H would be easier to read if genotypes were on the X axis. Overall the genetic data presented does not clarify if fat and faf act in a single pathway or in parallel. In this reviewer's opinion the FatGRV/Fd is likely a null, so the enhancement suggests a parallel pathway.

Response: We now include the genotypes in **Figure 2h** for further clarity. We agree with the reviewer that genetic interactions do not entirely clarify if *ft* and *faf* act in a single pathway or in parallel and we highlighted this in our interpretation of the results, as it is still possible that some of the effects are due to a Ft-independent role (Lines 151 and 152). We additionally assessed whether the trans-heterozygous *ft* combination resulted in total loss of Ft protein and whether loss of one copy of *faf* affected this phenotype. The new **Figure S3**

shows that, indeed as expected, the *ft* trans-heterozygous mutant combination has a dramatic decrease in Ft protein levels at the membrane (**Figure S3b**). Remarkably, loss of one copy of *faf^{β3}* resulted in a further decrease in Ft levels, in line with the observed phenotypes (**Figure S3c**). Whilst this does not prove that the observed enhancement of the phenotype is exclusively via *ft*, it suggests that Ft is involved.

4) Figure 3 shows more genetic interactions with Ds, Dlish and Fbxl7 generally support the concept that Faf interacts with the Fat-Hippo pathway.

Response: We thank the reviewer for their comment regarding the fact that this data supports our hypothesis.

5) Figure 4 is not very clear. Panel C is not clear – what are the bands on the left side of the gel? Not clear how I and J is different – quantification would help to be convincing of a change in overexpressed Fat protein levels upon *faf* depletion.

Response: We apologise for the lack of clarity on **Figure 4**. The bands on the left side of the gel in panel **c** were from other samples from the same cell-based experiment. We now include a new **Figure 4c** panel with a new repeat of the experiment showing that expression of *Faf^{LD}* promotes *Ft^{CD}* stabilisation. With regards to the data shown in **Figure 4i** and **4j**, the corresponding quantifications are presented in **Figure S6e-g**, showing there is a reduction in Ft protein levels when *faf* is depleted.

6) Figure 5 / D staining is difficult to see – quantification of levels and cytoplasmic/membrane ratios would help. Perhaps mosaics would make this clearer? Putting Armadillo in the middle makes it more difficult to compare directly the panels of D.

Response: We thank the reviewer for the comments on **Figure 5** and have updated the figure to improve clarity. The new version of **Figure 5** includes all genotypes analysed and the data pertaining to Armadillo has been moved to **Figure S7** to facilitate comparisons. Additionally, we now include plot profiles of Dachs and Armadillo levels showing the effect of specific genetic combinations on Dachs localisation (**Figure 5b''-g''**). We agree with the reviewer that mosaic analysis would have been ideal in this situation but, despite many attempts, unfortunately we were unable to generate transgenic stocks to generate *faf* mutant clones as we did not recover stocks with *faf* mutants combined with an appropriate *FRT*.

7) Figure 6. Overall, the changes in *ex-lacZ* in this figure are more clear than those of Diap-GFP. It is obvious that *faf* RNAi leads to increased *ex-lacZ*, and that overexpressed Ft decreases Hippo targets. Overexpression of Fat and knockdown of *faf* in the picture seems to be equal but in the quantification it is different. Perhaps the figure is not representative?

Response: Whilst the *ex-lacZ* results are clearer to visualise, these can sometimes be misleading in the absence of quantification. As we are quantifying the ratio between the levels of *ex-lacZ* in the posterior compartment (GFP-positive) and in the anterior compartment (GFP-negative), the ratio depends on the relative size of the compartments and, therefore, changes in the ratio can be masked if there are changes in the relative size of the compartments. For the *ft; faf^{RNAi}* samples, the image selected has a P/A ratio of 1.185, comparable to the average of all samples (1.151). We selected an image that reflects the average and does not mislead the effect of *faf* RNAi depletion in the context of *ft* over-expression. The main point we want to convey is that *faf* depletion abrogates the effect of *ft*, which is obvious when comparing **Figure 6d** and **Figure 6e**. Below, we show other examples of representative images of the *ft; faf^{RNAi}* genotype, with an indication of the respective *ex-lacZ* P/A ratio (**Rebuttal Figure 1**). We are happy to update the image in **Figure 6e** for the final version if necessary, but would prefer to not overstate the results obtained.

Original Figure 6e image

ex-lacZ P/A ratio

1.185

ex-lacZ P/A ratio

1.135

ex-lacZ P/A ratio

1.136

ex-lacZ P/A ratio

1.391

ex-lacZ P/A ratio

1.265

ex-lacZ P/A ratio

1.331

Rebuttal Figure 1 – Additional data related to Figure 6. XY confocal sections of third instar wing imaginal discs containing *ex-lacZ* (a-f, shown in red in a'-f' merged images), in which *en-Gal4* was used to drive expression of *UAS-ft* and *UAS-faf^{RNAI}*. *ex-lacZ* P/A ratio for each sample is indicated on the right. Note that a) is the original image shown in Figure 6e of the manuscript. GFP (green in a'-e' merged images) indicates posterior compartment where transgenes are expressed. DAPI (blue) stains nuclei. Dashed lines indicate boundary between anterior and posterior compartments.

8) Fig 7 The use of *Faf* and *faf* wt is confusing. Does *Faf*-wt express more transcript? qPCR could help clarify that, if this is an important point.

Response: We thank the reviewer for their suggestion. We have altered the nomenclature of the *faf* alleles for increased clarity, and they are now depicted as *faf^{isoC}* and *faf^{WT}*. We also show in **Figure S8s** that, indeed, it appears that *faf^{WT}* is expressed at a higher level than that of *faf^{isoC}*, by RT-PCR analysis.

9) Fig 8. Panel G shows *faf* stabilizes Fat. But is it 2x as shown in (H)? The quantification does not seem to reflect the gel shown.

Response: We thank the reviewer for their comment. The corresponding experiment was repeated, and we now show an updated **Figure 8g** in which the difference between the effect of WT Faf and its catalytic mutant version is much clearer. Quantification of this effect was also updated in **Figure 8h**.

10) Fig 9. Ex LacZ effect is clear but quantification of *diap1* is not clearly matching- perhaps the authors could comment on this?

Response: We apologise for not providing all the necessary information in **Figure 9**. The data accurately represents the quantification of the DIAP1::GFP levels, but this was perhaps unclear in the absence of the respective control sample. Contrary to *ex-lacZ*, in this set of experiments the ratio of *DIAP1::GFP* levels in the posterior and anterior compartment is not ~1 in controls and, therefore, the results shown in the original figure gave the impression that USP9X had no effect when, in fact, there was a reduction compared to the controls. To clarify this point, the new version of **Figure 9** includes the respective control (**Figure 9c**), which better reflects the quantification shown in **Figure 9f**.

11) It would strengthen the paper to show that Fat is ubiquitinated and to see if Faf affects the levels of Fat ubiquitination, in the S2 cell system. However, this is not essential. Similarly, to be confident that PCP is not affected the authors could examine wing hair orientation or ridges. Quantification of roundness could be in main figures.

Response: We agree with the reviewer in all the points listed above. First, we attempted to study the levels of Ft ubiquitylation in S2 cells and we show below a representative experiment of our efforts (**Rebuttal Figure 2**). As shown, in the absence of a specific E3 that promotes its degradation, the levels of Ft ubiquitylation are quite low, which makes it very difficult to assess whether Faf abrogates this. In one of our experiments, we observed a very significant effect of the catalytic mutant Faf, which resulted in an increase in Ft ubiquitylation levels, suggestive of a potential dominant-negative effect of Faf. However, this was not reproduced to the same extent in other independent repeats.

Rebuttal Figure 2 – Ft ubiquitylation assay. *Drosophila* S2 cells were transfected with the indicated constructs 48h prior to cell lysis. S2 cells were also additionally treated with the proteasome inhibitor MG132 4h before cell lysis. Following lysis under denaturing conditions, ubiquitylated proteins were isolated using anti-HA antibodies. The presence of Ft^{ICD} and Faf was assessed with the indicated antibodies. Note that expression of Faf catalytic mutant (Faf^{CD}) resulted in accumulation of ubiquitylated Ft^{ICD}, which was not visible in the control sample. Long exposure of HA immunoprecipitates probed with FLAG antibody appears to depict very low levels of Ft ubiquitylation in the control sample, which were not detected when Faf WT was expressed. ∅ and + indicate absence or presence of the indicated plasmid, respectively. Asterisks denote non-specific bands. Tubulin was used as loading control.

To address the reviewer's point that the effects on PCP are still unclear, we analysed wing hair orientation in adult wings in different genotypes (**Figure S2i**). This analysis did not show any overt effects of Faf on wing hair orientation. Moreover, we followed the reviewer's suggestion and now include the quantification of roundness (PD/AP axes ratio) in **Figure 1k**.

Reviewer #2 (Remarks to the Author):

This manuscript identified the deubiquitination enzyme Fat facets (Faf) as a positive regulator of Fat (Ft) function in controlling organ growth. Ft has been well characterized as an upstream activator of Hippo signalling, however its regulation by Faf has not been reported. This study fills this gap and advances our understanding of how ubiquitination controls Hpo signaling via the Faf/Ft axis. The experiments are logically and clearly presented, and the data are generally of high quality. Faf and Ft interact genetically and physically, and both regulate organ growth and Hpo target gene expression. The effects of Faf on Ft function require catalytic activity. The Faf/Ft interaction appears to be conserved, as the key interactions are recapitulated by using mammalian orthologs.

Response: We thank the reviewer for their comments and assessment of the strengths of our report.

The manuscript can be strengthened by addressing the following:

Major:

1) There were other DUBs identified in the initial screen (Fig. S1) that partially suppressed Ft-mediated wing undergrowth phenotype. Why was Faf selected for further study? Do the other DUBs also contribute to Ft regulation, i.e. does their perturbation also affect Ft function?

Response: The reviewer is right that there were other DUBs that were identified as partial suppressors of Ft-mediated wing phenotypes. Faf was selected as the focus of our study due to the fact that it had the strongest effect in the *in vivo* screen (**Figure S1c**), displayed an effect on tissue growth when depleted in the absence of Ft (**Figure S1b**), and its orthologue (USP9X) had been associated with the regulation of Hippo signalling in the mammalian setting. Based on the reviewer's observation, we assessed whether the other DUBs that were found as hits in the screen also contribute to Ft regulation (*Cyld* and *CG1503*, the orthologue of human SENP8). Remarkably, our preliminary analysis suggests that both *Cyld* and *CG1503* seem to regulate Ft protein levels in S2 cells and co-immunoprecipitate with Ft (**Rebuttal Figure 3**). We will actively investigate the underlying mechanism in future; however, given the current focus on Faf in this manuscript, we believe that the study of the role of *Cyld* and *CG1503* is better suited to a future report.

Rebuttal Figure 3 – Regulation of Ft protein levels and protein-protein interaction with *in vivo* DUB screen hits. *Drosophila* S2 cells were transfected with the indicated plasmids 48h prior to lysis and processing for FLAG immunoprecipitation or Western blot analysis. Co-IPs were performed with FLAG-tagged GFP, Faf^{LD}, Cyld (isoC or isoD) or CG1503 and HA-tagged Ft^{ICD}. The expression and presence of co-purified proteins were analysed by immunoblotting with the indicated antibodies. Tubulin was used as loading control. Note that all proteins tested were able to interact with Ft^{ICD} and regulate its protein levels.

2) Does *faf* genetically interact with Hippo pathway components other than Ft and its immediate network (e.g. Hpo, wts, yki, sd)? In other words, are the effects going through Hpo core components? This is important to ascertain in order to better place Faf in the pathway

Response: Based on the reviewer's suggestion, we assessed the epistatic position of Faf within the pathway. For this, we combined *faf* depletion or over-expression in the presence of *wts* over-expression or *hpo* RNAi-mediated depletion (**Figure S5**). This analysis revealed that *faf* is most likely acting upstream of the core kinase cascade as the phenotypes elicited by *hpo* depletion (**Figure S5a-d**) or *wts* expression (**Figure S5e-k**) were largely unaffected by the modulation of *faf* levels.

3) The story would benefit from a more rigorous demonstration of the effects of Faf on Ft protein levels. Since demonstration of decreased Ft ubiquitination in the presence of Faf appears technically challenging (as stated in Discussion), other molecular assays could be used. For example, at least the effects of proteasome inhibitors should be tested, and also Ft stability with and without Faf can be tested in a cycloheximide chase assay.

Response: We thank the reviewer for the suggestions to strengthen the effect of Faf-mediated regulation of Ft protein levels. We now include evidence that Ft is likely being degraded in a proteasome-dependent manner as MG132 treatment can rescue Ft levels (**Figure S9a,b**). Interestingly, Faf over-expression has a much stronger effect than MG132 (**Figure S9a**), which leaves open the possibility that proteasome-independent mechanisms may also be at play.

4) The specificity of the DUB inhibitor WP1130 should be described in the text. How specific is it for Faf? Is it possible that it inhibits other DUBs?

Response: The reviewer raises an important point. We mention in the manuscript text that it has been suggested that WP1130 inhibits other DUBs beyond Faf. Despite this, we believe that the effect of WP1130 is mostly due to its inhibition of Faf. Firstly, apart from Faf, none of the other predicted WP1130-inhibited DUBs were found as hits in the *in vivo* screen. Secondly, when we performed experiments combining WP1130 treatment with Faf RNAi-mediated depletion or over-expression in S2 cells (**Figure S9c**), we observed that the effect of WP1130 was largely attenuated, suggesting that other DUBs that are predicted to be affected by the inhibitor have a minor effect in this mechanism, if any.

Minor:

1) Was modulation of the L2 vein phenotype in Fig S2I significant? Statistical assessment should be provided.

Response: Based on the reviewer's comment, we now include in **Figure S2h** the statistical analysis associated with the L2 vein phenotype. Significance was calculated using Chi-square analysis, adjusted Fisher's exact test and Bonferroni correction for multiple comparisons and we can see an effect of modulating *faf* function in many of the genetic backgrounds analysed.

2) Effects on D (Fig 5 and other figures) are not particularly striking. Maybe use enlargements to show effects in single cells, and point out the differences?

Response: We thank the reviewer for this suggestion. As mentioned above in the response to **reviewer 1**, we now include plot profiles of Dachs and Armadillo, which more clearly show the effect of modulating *faf* levels.

3) Fig 5E panels are not referred to in the text.

Response: We apologise for this oversight. **Figure 5e** is now referred to when discussing the data in **Figure 5**.

4) Line 451, "Faf directly affected Hippo signalling readouts in vivo" – unclear what this means. There was an effect on Hpo target genes, but the word 'directly' should be dropped, because obviously Faf does not transcriptionally regulate Hpo reporter genes.

Response: We thank the reviewer for pointing out this error. This has been corrected in the revised version.

Reviewer #3 (Remarks to the Author):

The manuscript by Dawson et al., explores roles for ubiquitination in the Hippo signalling pathway in *Drosophila*, with a focus on the Fat cadherin branch of the pathway. They perform a small scale *in vivo* genetic screen and identify the deubiquitylating enzyme Fat facets (Faf) as a positive regulator of Fat signalling in the context of tissue growth control. This adds new information to Hippo/Fat signalling. In general, the study is high on technical rigor – i.e. the wing growth experiments, immunofluorescence and biochemistry. The data are mostly convincing, well-controlled and rigorous, although some alternative experimental approaches and further quantification are required to fully prove the reported role of Faf. This is important as some of the effects are quite subtle - e.g. the impact of Faf depletion on Fat abundance, Dachs localisation and Yki activity. The subtle nature of the impact of Faf on Fat is reflected in the fact that the authors build their case through mass action (ten figures and many experiments) as opposed to a smaller set of very convincing results. The experiments on human USP9X and Fat4 do not add much value to the paper especially because they do not further clarify the role of Fat homologues in Hippo signalling in mammals, which is still very controversial.

Response: We thank the reviewer for their assessment of the manuscript, for their comments and suggestions.

Major:

1) The major thing that is currently lacking from the manuscript is a precise understanding of the relative importance of ubiquitination to Fat/Hippo signalling and growth. More specifically:

- when and where does Fat facets, and ubiquitination more broadly, influence Fat/Hippo signalling in growing tissues?
- does it have a major impact or a minor impact on Fat/Hippo signalling?
- Does it happen at specific stages of development and/or on other timescales, like specific periods of the cell cycle?
- Or does ubiquitination have special roles when cells encounter specific cues?
- Alternatively, does ubiquitination help set baselines or thresholds of Fat/Hippo signalling that are then modulated by other things like mechanical forces, or something else?

If the above questions cannot be addressed experimentally, they should at least be discussed. Currently, the Discussion is very long and focussed on molecular aspects of Faf and Fat. A more rounded Discussion of Faf would enhance the manuscript; some space for this could be made by removing repetition at the start of the Discussion, which recaps the introduction and results sections.

Response: We agree with the reviewer that the questions that they have posed are very important to assess, but we believe that they fall outside the scope of the current report as it would be technically challenging to conclusively answer most of them, particularly when the identity of the E3 ligase of interest for this mechanism remains unknown. These are indeed very much the focus of our future research. We believe that ubiquitylation is likely to act as a switch that modulates Ft function not only by controlling its protein levels, but also by controlling specific Ft-mediated interactions. The fact that we primarily detect an effect on the tissue growth functions of Ft and not an effect on PCP would fit with our hypothesis. The reviewer's suggestion that there may be a link with the response to mechanical forces is an attractive option and we plan to generate and/or exploit existing models to address this.

Based on the reviewer's comments, we have altered the Discussion section of the manuscript and removed most of the content that pertained to points covered in the Introduction and Results sections and now reference and discuss some of the points raised above.

Minor:

Other things

1) Figure 2. I may have missed it, but I could not find information on the disc size experiments in the results, legends or methods. Specifically, were the animals carefully staged and dissected at the same age? This is essential because fat loss substantially extends the larval period of development for several days and thus gives imaginal discs an extended growth period. Different results are possible depending on when the tissues are removed and analysed.

Response: We apologise for this oversight. We now include a section in the Materials and Methods where we detail how these experiments were performed. Briefly, crosses were setup at 25°C and flies were flipped to new vials every 24h for precise staging of larval development. Larvae were dissected at the late L3 stage (wandering stage), approximately 5 days after egg laying. Larvae from different genotypes were dissected at the same time to avoid differences associated with developmental delays due to loss of *ft* function.

2) Figure 4. The Fat abundance result in Figure 4 is very important and looks to be very modestly different if at all in 4F with Faf RNAi. This important result should be validated with *faf* mutant allele clones and high resolution/high magnification images should be presented if the result is consistent.

Also, the current data here should be quantified across multiple tissues and high-resolution images should be presented in the main figure or the supp figures. This is done for the USP9X experiments in Fig 10 but not in Fig 4.

Finally, what is the prominent D/V band in 4F and 4H? This looks very different to the Fat stains in other figure panels and in published studies. It looks like Wingless protein and is distracting.

Response: We agree with the reviewer that the results in **Figure 4** are very important. We now include a new **Figure 4f** panel that hopefully reflects more accurately the changes in Ft protein levels associated with *faf* depletion. As mentioned above in the reply to **reviewer 1**, we were unsuccessful in generating transgenic stocks that would allow us to create *faf* mutant clones. For unknown reasons, despite multiple attempts, we did not recover any recombinant stocks with *faf* mutations associated with the appropriate *FRT* for mitotic clone generation. We agree that this would be ideal to provide further evidence of the effect of Faf in this process.

We apologise for the lack of clarity regarding the quantification of the results shown in **Figure 4**. These are provided in **Figure S6e-g**. Moreover, regarding the reviewer's comment on the effect of Faf in other tissues, we now include evidence that Faf can regulate Ft protein levels in the eye imaginal disc using *mirr-Gal4* as a driver (**Figure S6h-k**). Interestingly, it appears that the function of Faf is more prominent in the wing disc as only the expression of Faf, but not its RNAi-mediated depletion, resulted in significant changes in Ft protein levels.

Finally, regarding the prominent D/V band in the original **Figure 4f** and **4h**, we suspect that this may be due to the genetic background of the flies used and their floating balancers. To avoid confusion, we now provide new images for **Figure 4f** and **Figure 4h**.

3) Figure 5. The Dachs data are very difficult to follow. First, the inclusion of the 4 different gray panels per row that look very similar does not make it easy for readers to know what they are looking at. Second, the reported changes in Dachs are not readily appreciable by eye and appear to be very subtle. Dachs polarisation is easier to appreciate when done in a clonal fashion – e.g., PMID: 18694569. Can this clonal approach be used here to test Faf's role in Dachs localisation more convincingly? Alternatively, the Arm data could be removed to supps and high resolution images of Dachs shown. Is there any quantification?

The impact of Faf on Fat/Dachs could be further validated by exploring Vamana localisation, which is also sensitive to Fat/Dachs levels.

Response: We thank the reviewer for their comments, which are also in line with the comments from other reviewers. As mentioned above, we have updated **Figure 5** to improve clarity. The new version of **Figure 5** includes all genotypes analysed and the data pertaining to Armadillo has been moved to **Figure S7** to facilitate comparisons between the different genotypes, with regards to D localisation. We also include plot

profiles of Dachs and Armadillo levels showing the effect of specific genetic combinations on Dachs localisation (**Figure 5b''-g''**).

We agree with the reviewer's comment that assessment of D localisation in clones would have been ideal in this situation, but this proved unfeasible due to the reasons pointed above in the reply to the previous point. Finally, regarding the potential effect on Dlish/Vamana localisation, we were unable to source reagents for investigating this possibility *in vivo*. However, we analysed in S2 cells whether Faf affected Dlish protein levels. As shown in **Rebuttal Figure 4**, Faf did not affect Dlish protein levels in S2 cells. This suggests that the effect of Faf may not be due to a direct action on Dlish, although it remains possible that results could differ *in vivo* due to the fact that S2 cells may lack some of the relevant Ft signalling components. In addition, we also observed that Faf did not regulate the protein levels of Elgi (**Rebuttal Figure 5**).

Rebuttal Figure 4 – Dlish protein levels are not regulated by Faf in *Drosophila* S2 cells. S2 cells were transfected with the indicated plasmids 48h prior to lysis and processing for Western blot analysis with the indicated antibodies. GFP FLAG and Tubulin (Tub) were used as transfection and loading controls, respectively.

Rebuttal Figure 5 – Elgi protein levels are unaffected by Faf in *Drosophila* S2 cells. S2 cells were transfected with the indicated plasmids 48h prior to lysis and immunoblotting with the indicated antibodies. GFP FLAG and Tubulin (Tub) were used as transfection and loading controls, respectively.

3) Figure 6. Assessment of Yki activity is another important experiment and currently relies on RNAi. It should also be performed in *faf* mutant allele clones, especially because the RNAi results show significant elevation one Yki target gene but not the other. Also, looking at the data it appears that there are region-specific changes in Yki target genes. This could be teased out further with the mutant clone approach.

Response: We agree that assessing Yki activity in *faf* mutant clones is important, but due to the technical issues described above, we have been unable to address this directly. We respectfully disagree with the reviewer regarding the statement that Faf only regulates one of the Yki target genes. We have shown in **Figure 6**, **Figure 7**, **Figure 9** and **Figure S8** that modulating Faf levels and function regulates *ex-lacZ* and *DIAP1::GFP* expression.

With regards to the potential region-specific changes in the regulation of Yki target genes, we agree with the reviewer that this is indeed something very interesting. For unknown reasons, several Hpo pathway regulators appear to have a much more prominent effect in the wing pouch than in other regions of the wing

imaginal disc (e.g. Ck1alpha (PMID: 31567070) and Crumbs (PMID: 24778256)) and this phenomenon has been insufficiently studied. In the case of Faf, this could be due to the fact that Faf is associated with other signalling pathways and, perhaps the regulation of Ft is more prominent in regions where there is co-regulation of Hpo signalling and other pathways with which it crosstalks. This is something that needs to be further addressed in future experiments.

Based on the reviewers' suggestions, we performed the following changes:

1) Data present in the original **Figure 5** was moved to Supplementary information (new **Figure S8**) and conclusions regarding D polarisation have been toned down throughout the manuscript. Data regarding in vivo D localisation has been moved to new **Figure S9** and **Figure S10**;

2) Information regarding statistical analyses and n numbers has been included throughout;

3) Information on p-values has been compiled in **Supplementary Table 2**.;

4) **Figure 7** and **Figure 9** include plot profiles for D and Arm localisation to highlight differences;

5) Original **Figure 6** is now new **Figure 5**;

6) Original Figure 7 is now new **Figure 6** and includes information previously included in original **Figure S8**;

7) Data in original **Figure S6** is now shown in new **Figure S6** and new **Figure S7**;

8) Figure legends have been revised and shortened;

9) Source data for all main Figures has been compiled and provided as a Source data file;

10) Source data for all Supplementary Figures has been compiled and provided as a Source data file;

11) In accordance with the points above, we performed extensive revisions of the manuscript text.